# Latent Representation and Simulation of Markov Processes via Time-Lagged Information Bottleneck

**Marco Federici**[*][†]
AMLab
University of Amsterdam
m.federici@uva.nl

**Patrick Forré**
AI4Science Lab, AMLab
University of Amsterdam
p.d.forre@uva.nl

**Ryota Tomioka**
Microsoft Research AI4Science
ryoto@microsoft.com

**Bastiaan S. Veeling**[*]
Microsoft Research AI4Science
basveeling@microsoft.com

## Abstract

Markov processes are widely used mathematical models for describing dynamic systems in various fields. However, accurately simulating large-scale systems at long time scales is computationally expensive due to the short time steps required for accurate integration. In this paper, we introduce an inference process that maps complex systems into a simplified representational space and models large jumps in time. To achieve this, we propose Time-lagged Information Bottleneck (T-IB), a principled objective rooted in information theory, which aims to capture relevant temporal features while discarding high-frequency information to simplify the simulation task and minimize the inference error. Our experiments demonstrate that T-IB learns information-optimal representations for accurately modeling the statistical properties and dynamics of the original process at a selected time lag, outperforming existing time-lagged dimensionality reduction methods.

## 1 Introduction

Markov processes have long been studied in the literature (Norris, 1997; Ethier & Kurtz, 2009), as they describe relevant processes in nature such as weather, particle physics, and molecular dynamics. Despite being well-understood, simulating large systems over extensive timescales remains a challenging task. In molecular systems, analyzing meta-stable molecular configurations requires unfolding simulations over several milliseconds ($\tau \approx 10^{-3}s$), while accurate simulation necessitates integration steps on the order of femtoseconds ($\tau_0 \approx 10^{-15}s$). The time required to simulate $10^{12}$ steps is determined by the time of a single matrix multiplication, which takes on the order of milliseconds on modern hardware, resulting in a simulation time of multiple years.

Deep learning-based approximations have shown promising results in the context of time series forecasting (Staudemeyer & Morris, 2019; Lim & Zohren, 2021), including applications in weather forecasting (Veillette et al., 2020), sea surface temperature prediction (Ham et al., 2019; Gao et al., 2022), and molecular dynamics (Sidky et al., 2020; Klein et al., 2023; Schreiner et al., 2023). Mapping observations into lower-dimensional spaces has proven to be an effective method for reducing computational costs. Successful examples in molecular dynamics include learning system dynamics through coarse-grained molecular representations (Wang et al., 2019a; Köhler et al., 2023; Arts et al., 2023), or linear (Koopman, 1931; Molgedey & Schuster, 1994) and non-linear (Wehmeyer & Noé, 2018; Mardt et al., 2018; Sidky et al., 2020) projections of molecular features.

Modern deep representation learning methods have proven effective in creating representations for high-dimensional structured data, including images (Hjelm et al., 2019; Chen et al., 2020), audio

---

[*]Corresponding author.
[†]Work partially done during an internship at Microsoft Research, AI4Science.

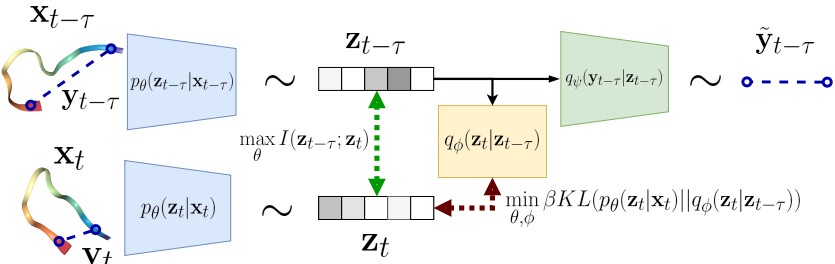

Figure 1: The Time-lagged Information Bottleneck objective aims to maximize the mutual information between sampled representations $\mathbf{z}_{t-\tau}, \mathbf{z}_t$ at temporal distance $\tau$ while minimizing mismatch between the encoding distribution $p_\theta(\mathbf{z}_t|\mathbf{x}_t)$ and the learned variational transitional distribution $q_\phi(\mathbf{z}_t|\mathbf{z}_{t-\tau})$. This results in minimal representations capturing dynamics at timescale $\tau$ or larger, which can be used to predict properties of interest $\mathbf{y}_t$, such as inter-atomic distances, over time.

(van den Oord et al., 2018; Saeed et al., 2021), text (Devlin et al., 2018; Radford et al., 2018), and graphs (Veličković et al., 2018; Wang et al., 2022). These methods often aim to capture relevant information while reducing the complexity of the data. In this context, information theory provides a compelling direction for further analysis (Wennekers & Ay, 2003; Gao et al., 2022; Lozano-Durán & Arranz, 2022). In particular, the information bottleneck principle (Tishby et al., 2000; Tishby & Zaslavsky, 2015) suggests that an optimal representation should retain relevant information while discarding unnecessary features. Applying this principle to the context of Markov process simulations has the potential to simplify the modeling task, reduce computational complexity, and aid in identifying the salient characteristics that define the relevant dynamics.

In this paper, we make the following contributions: (i) we introduce a probabilistic inference scheme for Markov processes, *Latent Simulation* (LS), and characterize the inference error by defining *Time-lagged InfoMax* (T-InfoMax) as a general family of principled training objectives. (ii) We propose *Time-lagged Information Bottleneck* (T-IB, Figure 1), a novel objective that follows the T-InfoMax principle to preserve system dynamics while discarding superfluous information to simplify modeling tasks. (iii) We empirically compare the performance of models trained using the T-InfoMax and T-IB objectives on synthetic trajectories and molecular simulations, showcasing the importance of the T-InfoMax principle and the advantages of the proposed T-IB method for both representation learning and latent simulation inference compared to other models in the literature.

## 2 METHOD

We delve into the problem of efficiently representing and simulating Markov processes starting by defining *Latent Simulation* as an inference procedure and characterizing the corresponding error (section 2.1). Next, in section 2.2, we analyze the problem of capturing system dynamics from an information-theoretic perspective, defining and motivating *Time-Lagged InfoMax*: a family of objectives that minimizes the latent simulation error. Finally, we introduce *Time-lagged Information Bottleneck* (section 2.3) as an extension of T-InfoMax that aims to simplify the representation space. A schematic representation of our proposed model is visualized in Figure 1.

### 2.1 LATENT SIMULATION

Consider a sequence of $T$ random variables, denoted as $[\mathbf{x}_t]_{t=0}^T$, which form a homogeneous Markov Chain. This chain models a dynamical process of interest, such as molecular dynamics, global climate systems, or particle interactions. Let $\mathbf{y}_t$ represent a specific (noisy) property of $\mathbf{x}_t$ that we aim to model over time. Formally, we define $\mathbf{y}_t = f(\mathbf{x}_t, \epsilon_t)$, where $f : \mathbb{X} \times \mathcal{E} \to \mathbb{Y}$ is some function and $\epsilon_t$ is temporally uncorrelated noise. Examples of such properties could include the energy or momentum of a particle, the meta-stable state of a molecular structure, and the amount of rainfall. Each of these properties $\mathbf{y}_t$ can be derived from a more comprehensive high-dimensional state description $\mathbf{x}_t$.

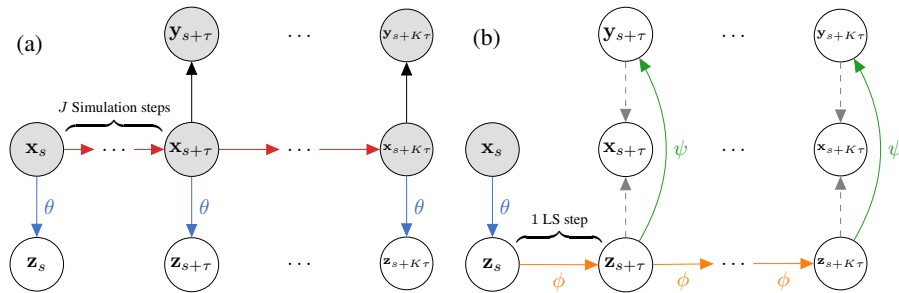

Figure 2: Graphical models for the joint distribution of a sequence, targets, and representations. 2a shows the data-generating process in which red arrows denote computationally expensive simulation steps. 2b represents the corresponding Variational Latent Simulation, in which the transitions are modeled in the latent space. Gray dashed lines indicate distributions that are not used for inference.

Given an initial observation $\mathbf{x}_s$, the joint distribution of the sequence of $K$ future targets $[\mathbf{y}_{s+k\tau}]_{k=1}^K$ at some *lag time* $\tau > 0$ can be expressed as:

$$p([\mathbf{y}_{s+k\tau}]_{k=1}^K \,|\, \mathbf{x}_s) = \int \ldots \int \prod_{k=1}^K \underbrace{p(\mathbf{x}_{s+k\tau}|\mathbf{x}_{s+(k-1)\tau})}_{\text{Transition}} \underbrace{p(\mathbf{y}_{s+k\tau}|\mathbf{x}_{s+k\tau})}_{\text{Prediction}} \, d\mathbf{x}_{s+\tau} \ldots d\mathbf{x}_{s+K\tau}. \quad (1)$$

Each transition distribution $p(\mathbf{x}_{t+\tau}|\mathbf{x}_t)$ may necessitate $J$ integration steps at a finer timescale $\tau_0 < \tau$. Given the sequential nature of simulation, generating trajectories over extended time horizons may require substantial computational resources. To mitigate the challenges of simulating large-scale system dynamics, we adopt two modeling strategies: (i) rather than modeling the transition distribution in the original space $\mathbb{X}$, we learn a time-independent *encoder* $p_\theta(\mathbf{z}_t|\mathbf{x}_t)$ that maps into a simpler representation space $\mathbb{Z}$; and (ii) we directly model the dynamics for larger jumps $\tau > \tau_0$. We refer to the process of unfolding simulations in the latent representation space as *Latent Simulation* (LS). The joint distribution for trajectories of targets unfolded using LS starting from $\mathbf{x}_s$ is defined as:

$$p^{LS}([\mathbf{y}_{s+k\tau}]_{k=1}^K \,|\, \mathbf{x}_s) := \int \ldots \int \underbrace{p_\theta(\mathbf{z}_s|\mathbf{x}_s)}_{\text{Encoding}} \prod_{k=1}^K \underbrace{p(\mathbf{z}_{s+k\tau}|\mathbf{z}_{s+(k-1)\tau})}_{\text{Latent transition}} \underbrace{p(\mathbf{y}_{s+k\tau}|\mathbf{z}_{s+k\tau})}_{\text{Latent prediction}} \, d\mathbf{z}_s \ldots \mathbf{z}_{s+K\tau}. \quad (2)$$

Unfolding LS requires access to the *latent transition* $p(\mathbf{z}_{t+\tau}|\mathbf{z}_t)$ and *predictive* $p(\mathbf{y}_t|\mathbf{z}_t)$ distributions, which are generally intractable for an arbitrary choice of encoding distribution $p_\theta(\mathbf{z}_t|\mathbf{x}_t)$. To circumvent this intractability, we introduce *variational transition* and *variational target predictive* distributions, denoted as $q_\phi(\mathbf{z}_t|\mathbf{z}_{t-\tau})$ and $q_\psi(\mathbf{y}_t|\mathbf{z}_t)$, respectively. The resulting joint inference distribution for the future targets $[\mathbf{y}_{s+k\tau}]_{k=1}^K$, unfolding from the initial observation $\mathbf{x}_s$, is referred to as the *Variational Latent Simulation* distribution $q^{LS}([\mathbf{y}_{s+k\tau}]_{k=1}^K \,|\, \mathbf{x}_s)$ and visualized together with the graphical model for the data-generating process in Figure 2.

The Kullback-Leibler (KL) divergence, which quantifies the discrepancy between the ground truth and the variational latent simulation distributions, can be upper-bounded as follows:

$$\underbrace{\text{KL}(p([\mathbf{y}_{s+k\tau}]_{k=1}^K \,|\, \mathbf{x}_s) \| q^{LS}([\mathbf{y}_{s+k\tau}]_{k=1}^K \,|\, \mathbf{x}_s))}_{\text{Variational Latent Simulation error}} \leq \underbrace{\text{KL}(p([\mathbf{y}_{s+k\tau}]_{k=1}^K \,|\, \mathbf{x}_s) \| p^{LS}([\mathbf{y}_{s+k\tau}]_{k=1}^K \,|\, \mathbf{x}_s))}_{\text{Latent Simulation error}}$$

$$+ \sum_{k=1}^K \underbrace{\text{KL}(p(\mathbf{z}_{s+k\tau}|\mathbf{z}_{s+(k-1)\tau}) \| q_\phi(\mathbf{z}_{s+k\tau}|\mathbf{z}_{s+(k-1)\tau}))}_{\text{Variational latent transition gap}} + \underbrace{\text{KL}(p(\mathbf{y}_{s+k\tau}|\mathbf{z}_{s+k\tau}) \| q_\psi(\mathbf{y}_{s+k\tau}|\mathbf{z}_{s+k\tau}))}_{\text{Variational latent prediction gap}}. \quad (3)$$

The upper bound consists of the latent simulation error and the sum of the variational gaps for both the latent transition and target predictive distributions. Unfortunately, terms on the right side of equation 3 are intractable. To address this, we propose a two-step optimization procedure: (i) we first learn an encoding distribution $p_\theta(\mathbf{z}_t|\mathbf{x}_t)$ that minimizes the latent simulation error, effectively capturing the dynamical properties of the system in the representation; then (ii), assuming a fixed (optimal) encoding distribution, we optimize the variational latent transition $q_\phi(\mathbf{z}_{t+\tau}|\mathbf{z}_t)$ and

predictive $q_\psi(\mathbf{y}_t|\mathbf{z}_t)$ distributions using maximum likelihood to minimize their respective variational gaps. In the following sections, we will focus on step (i), analyzing the problem of learning representations that preserve dynamical properties from an information theoretical perspective. Additional details about this two-step procedure are available in Appendix C.1.

## 2.2 TEMPORAL INFORMATION ON MARKOV CHAINS

A crucial prerequisite for ensuring that the latent simulation process does not introduce any error is to guarantee that each representation $\mathbf{z}_t$ is as informative as the original data $\mathbf{x}_t$ for the prediction of any future target of interest $\mathbf{y}_{t+\tau}$. If $\mathbf{z}_t$ is less predictive than $\mathbf{x}_t$ for $\mathbf{y}_{t+\tau}$, the statistics for the corresponding predictive distribution $p(\mathbf{y}_{t+\tau}|\mathbf{z}_t)$ would deviate from those based on the original data $p(\mathbf{y}_{t+\tau}|\mathbf{x}_t)$. This first requirement can be expressed by equating *mutual information*[1] that $\mathbf{x}_t$ and $\mathbf{z}_t$ share with $\mathbf{y}_{t+\tau}$: $I(\mathbf{x}_t; \mathbf{y}_{t+\tau}) = I(\mathbf{z}_t; \mathbf{y}_{t+\tau})$. We will refer to this requirement as *sufficiency* of $\mathbf{z}_t$ for $\mathbf{y}_{t+\tau}$. Sufficiency is achieved only when $\mathbf{x}_t$ and $\mathbf{z}_t$ yield identical predictive distributions for the future target, i.e., $p(\mathbf{y}_{t+\tau}|\mathbf{x}_t) = p(\mathbf{y}_{t+\tau}|\mathbf{z}_t)$.

Secondly, we introduce the concept of *autoinformation*. Autoinformation at a given lag time $\tau$ is defined as the mutual information between the current observation $\mathbf{x}_t$ and its corresponding future $\mathbf{x}_{t+\tau}$. Formally, $AI(\mathbf{x}_t; \tau) := I(\mathbf{x}_t; \mathbf{x}_{t+\tau})$. This concept extends the statistical notion of autocorrelation, which measures the linear relationship between values of a variable at different times (Brockwell & Davis, 2002), to include nonlinear relationships (Chapeau-Blondeau, 2007; von Wegner et al., 2017).

Since $\mathbf{z}_t$ is derived from $\mathbf{x}_t$, the autoinformation for $\mathbf{x}_t$ sets an upper-bound for the autoinformation for $\mathbf{z}_t$: $AI(\mathbf{x}_t; \tau) \geq AI(\mathbf{z}_t; \tau)$ (proof in Appendix B.3). We refer to the difference between the two values as the *autoinformation gap* $AIG(\mathbf{z}_t; \tau) := AI(\mathbf{x}_t; \tau) - AI(\mathbf{z}_t; \tau)$ and we say that $\mathbf{z}_t$ *preserves autoinformation* whenever autoinformation gap is zero.

**Lemma 1.** *Autoinformation and Sufficiency (proof in Appendix B.5)*
*A representation $\mathbf{z}_t$ preserves autoinformation at lag time $\tau$ if and only if it is sufficient for any target $\mathbf{y}_{t+\tau}$. Conversely, whenever $\mathbf{z}_t$ does not preserve autoinformation for a lag time $\tau$ is always possible to find a target $\mathbf{y}_{t+\tau}$ for which $\mathbf{z}_t$ is not sufficient:*

$$AIG(\mathbf{z}_t; \tau) = 0 \iff I(\mathbf{x}_t; \mathbf{y}_{t+\tau}) = I(\mathbf{z}_t; \mathbf{y}_{t+\tau}) \quad \forall \mathbf{y}_{t+\tau} := f(\mathbf{x}_{t+\tau}, \epsilon).$$

In simpler terms, a representation that preserves autoinformation encapsulates all dynamic properties of the original data for the temporal scale $\tau$. As a result, the representation $\mathbf{z}_t$ can replace $\mathbf{x}_t$ in predicting any future properties at time $t + \tau$.

For a temporal sequence $[\mathbf{x}_t]_{t=s}^T$, we define the autoinformation at lag time $\tau$ as the average autoinformation between all pairs of elements in the sequence that are $\tau$ time-steps apart: $AI([\mathbf{x}_t]_{t=s}^T; \tau) := \mathbb{E}_{t \sim U(s, T-\tau)} [AI(\mathbf{x}_t; \tau)]$, where $U(s, T - \tau)$ refers to a uniform distribution. If $p(\mathbf{x}_s)$ is stationary, the amount of autoinformation for a sequence $[\mathbf{x}_t]_{t=s}^T$ is equivalent to autoinformation at any point $\mathbf{x}_t$. Using this definition, we can show:

**Lemma 2.** *Autoinformation and Markov Property (proof in Appendix B.6)*
*If a sequence of representations $[\mathbf{z}_t]_{t=s}^T$ of a homogeneous Markov chain $[\mathbf{x}_t]_{t=s}^T$ preserves autoinformation at lag time $\tau$, then any of its sub-sequences of elements separated by $\tau$ time-steps must also form a homogeneous Markov chain:*

$$AIG([\mathbf{z}_t]_{t=s}^T; \tau) = 0 \implies [\mathbf{z}_{s'+k\tau}]_{k=0}^K \text{ is a homogeneous Markov Chain,}$$

*for every $s' \in [s, T - \tau]$, $K \in [0, \lfloor (T - s')/\tau \rfloor]$.*

Building on this, we further establish that dynamics at a predefined timescale $\tau$ also encode information relevant to larger timescales:

**Lemma 3.** *Slower Information Preservation (proof in Appendix B.8)*
*Any sequence of representations $[\mathbf{z}_t]_{t=s}^T$ that preserves autoinformation at lag time $\tau$ must also preserve autoinformation at any larger timescale $\tau'$:*

$$AIG([\mathbf{z}_t]_{t=s}^T; \tau) = 0 \implies AIG([\mathbf{z}_t]_{t=s}^T; \tau') = 0 \quad \forall \tau' \geq \tau.$$

---

[1]We refer the reader to Appendix A for further details on the notation.

By synthesizing the insights from Lemma 1, 2, and 3, we can infer that any representation preserving autoinformation at lag time $\tau$ captures the dynamical properties of the system across timescales $\tau'$ that are equal or larger than $\tau$. Specifically, we conclude that: (i) $\mathbf{z}_t$ can replace $\mathbf{x}_t$ in predicting any $\mathbf{y}_{t+\tau'}$ (Lemma 1 + Lemma 3); (ii) any sequence of representations $[\mathbf{z}_{s+k\tau'}]_{k=0}^{K}$ will form a homogeneous Markov Chain (Lemma 2 + Lemma 3). Furthermore, we establish an upper bound for the expected Latent Simulation error in equation 3 using the autoinformation gap:

$$\mathbb{E}_t \underbrace{\left[ \mathrm{KL}(p([\mathbf{y}_{t+k\tau}]_{k=1}^{K} | \mathbf{x}_t) || p^{LS}([\mathbf{y}_{t+k\tau}]_{k=1}^{K} | \mathbf{x}_t)) \right]}_{\text{Latent Simulation error for } K \text{ simulations steps with lag time } \tau} \leq K \underbrace{AIG([\mathbf{z}_t]_{t=s}^{T}; \tau)}_{\text{Autoinformation gap for lag time } \tau} , \qquad (4)$$

with $t \sim U(s, s+\tau-1)$ and $T := s + (K+1)\tau - 1$. In words, the latent simulation error is upper-bounded by the product of the number of simulation steps and the autoinformation gap. A full derivation is reported in Appendix B.9.

Given that the autoinformation between elements of the original sequence is fixed, we can train representations that minimize the autoinformation gap at resolution $\tau$ by maximizing the autoinformation between the corresponding representations at the same or higher temporal resolution. We refer to this training objective as *Time-lagged InfoMax* (T-InfoMax):

$$\mathcal{L}^{\text{T-InfoMax}}([\mathbf{x}_t]_{t=s}^{T}, \tau; \theta) := AIG([\mathbf{z}_t]_{t=s}^{T}; \tau) = -\mathbb{E}_{t \sim U(s, T-\tau)}[I(\mathbf{z}_t; \mathbf{z}_{t+\tau})]. \qquad (5)$$

Among the various differentiable methods for maximizing mutual information in the literature (Poole et al., 2019; Hjelm et al., 2019; Song & Ermon, 2020), we focus on noise contrastive methods (InfoNCE) due to their flexibility and computational efficiency (van den Oord et al., 2018; Chen et al., 2020). Therefore, we introduce an additional *critic* architecture $F_\xi : \mathbb{Z} \times \mathbb{Z} \to \mathbb{R}$ with parameters $\xi$ to define an upper-bound on the T-InfoMax loss:

$$\mathcal{L}^{\text{T-InfoMax}}([\mathbf{x}_t]_{t=s}^{T}, \tau; \theta) \leq \mathcal{L}^{\text{T-InfoMax}}_{\text{InfoNCE}}([\mathbf{x}_t]_{t=s}^{T}, \tau; \theta, \xi) \approx -\frac{1}{B} \sum_{i=1}^{B} \log \frac{e^{F_\xi(\boldsymbol{z}_{t_i}, \boldsymbol{z}_{t_i-\tau})}}{\frac{1}{B} \sum_{j=1}^{B} e^{F_\xi(\boldsymbol{z}_{t_j}, \boldsymbol{z}_{t_i-\tau})}}. \qquad (6)$$

In this equation, $t_i$ is sampled uniformly in the interval $(s, T-\tau)$, $\boldsymbol{z}_{t_i}$ and $\boldsymbol{z}_{t_i-\tau}$ are the representations of $\boldsymbol{x}_{t_i}$ and $\boldsymbol{x}_{t_i-\tau}$ encoded via $p_\theta(\mathbf{z}_t | \mathbf{x}_t)$, and $B$ denotes the mini-batch size. We refer the reader to Appendix C.2 for additional discussion regarding the proposed approximations.

## 2.3 FROM TIME-LAGGED INFOMAX TO TIME-LAGGED INFORMATION BOTTLENECK

In the previous section, we emphasized the importance of maximizing autoinformation for accurate latent simulation. However, it is also critical to design representations that discard as much irrelevant information as possible. This principle, known as *Information Bottleneck* (Tishby et al., 2000), aims to simplify the implied transition $p(\mathbf{z}_t | \mathbf{z}_{t-\tau})$ and predictive $p(\mathbf{y}_t | \mathbf{z}_t)$ distributions to ease the variational fitting tasks, decreasing their sample complexity. In dynamical systems, the information that $\mathbf{z}_t$ retains about $\mathbf{x}_t$ can be decomposed into the autoinformation at the lag time $\tau$ and superfluous information:

$$\underbrace{I(\mathbf{x}_t; \mathbf{z}_t)}_{\text{Total Information}} = \underbrace{AI(\mathbf{z}_{t-\tau}; \tau)}_{\text{Autoinformation at lag time } \tau} + \underbrace{I(\mathbf{x}_t; \mathbf{z}_t | \mathbf{z}_{t-\tau})}_{\text{Superfluous information}} . \qquad (7)$$

As shown in Appendix B.11, superfluous information consists of time-independent features and dynamic information for temporal scales smaller than $\tau$.

Incorporating sufficiency from equation 4 with the minimality of superfluous information we obtain a family of objectives that we denote as *Time-lagged Information Bottleneck* (T-IB):

$$\mathcal{L}^{\text{T-IB}}([\mathbf{x}_t]_{t=s}^{T}, \tau, \beta; \theta) = \mathcal{L}^{\text{T-InfoMax}}([\mathbf{x}_t]_{t=s}^{T}, \tau; \theta) + \beta \, \mathbb{E}_t \left[ I(\mathbf{x}_t; \mathbf{z}_t | \mathbf{z}_{t-\tau}) \right]. \qquad (8)$$

Here, $\beta$ is a hyperparameter that trades off sufficiency (maximal autoinformation, $\beta \to 0$) and minimality (minimal superfluous information, $\beta \to +\infty$). Given that superfluous information can not be computed directly, we provide a tractable upper bound based on the variational latent transition distribution $q_\phi(\mathbf{z}_t | \mathbf{z}_{t-\tau})$. Together with equation 6, this defines a tractable T-IB InfoNCE objective:

$$\mathcal{L}^{\text{T-IB}}_{\text{InfoNCE}}([\mathbf{x}_t]_{t=s}^{T}, \tau, \beta; \theta, \phi, \xi) \approx \frac{1}{B} \sum_{i=1}^{B} -\log \frac{e^{F_\xi(\boldsymbol{z}_{t_i}, \boldsymbol{z}_{t_i-\tau})}}{\frac{1}{B} \sum_{j=1}^{B} e^{F_\xi(\boldsymbol{z}_{t_j}, \boldsymbol{z}_{t_i-\tau})}} + \beta \log \frac{p_\theta(\boldsymbol{z}_{t_i} | \boldsymbol{x}_{t_i})}{q_\phi(\boldsymbol{z}_{t_i} | \boldsymbol{z}_{t_i-\tau})}, \qquad (9)$$

in which the encoder $p_\theta(\mathbf{z}_t | \mathbf{x}_t)$ is parametrized using a Normal distribution with learnable mean and standard deviation as in Alemi et al. (2016); Federici et al. (2020). Details on the upper bound in equation 9 are reported in Appendix C.3.

## 3 RELATED WORK

Information-theoretic methods have gained traction in fluid mechanics, offering valuable insights into energy transfer mechanisms (Betchov, 1964; Cerbus & Goldburg, 2013; Lozano-Durán & Arranz, 2022). Measures like *Transfer Entropy* (Schreiber, 2000) and *Delayed Mutual Information* (Materassi et al., 2014) closely align with the concept of *Autoinformation*, which is central in this work. However, previous literature predominantly focused on designing localized reduced-order models (Lozano-Durán & Arranz, 2022) by factorizing spatial scales and independent sub-system components, rather than learning flexible representations that capture dynamics at the desired temporal scale. Moreover, the theory and application of these principles have largely been confined to discrete-state systems (Kaiser & Schreiber, 2002) and model selection tasks (Akaike, 1974; Burnham & Anderson, 2004).

A widely used approach in dynamical system representation involves measuring and maximizing linear autocorrelation (Calhoun et al., 2001; Pérez-Hernández et al., 2013; Wiskott & Sejnowski, 2002). In particular, Sidky et al. (2020) proposes a latent simulation inference that leverages linear correlation maximization, coupled with a mixture distribution for latent transitions. As shown in Appendix D.1, autocorrelation maximization can be also interpreted as autoinformation maximization constrained to jointly Normal random variables (Borga, 2001). However, the linear restriction requires high-dimensional embeddings (Kantz & Schreiber, 2003; von Wegner et al., 2017), and may introduce training instabilities for non-linear encoders (Mardt et al., 2018; Wu & Noé, 2020; Lyu et al., 2022). In this work, we prove that the requirement of linear transitions is not necessary to capture slow-varying signals, demonstrating the benefits of modern non-linear mutual information maximization strategies.

The proposed T-InfoMax family also generalizes existing models based on the reconstruction of future states (Wehmeyer & Noé, 2018; Hernández et al., 2018). On one hand, these approaches are proven to maximize mutual information (Barber & Agakov, 2003; Poole et al., 2019), on the other their effectiveness and training costs are contingent on the flexibility of the decoder architectures (Chen et al., 2019). For this reason, we chose to maximize autoinformation using contrastive methods, which rely on a more flexible critic architecture (van den Oord et al., 2018; Hjelm et al., 2019; Chen et al., 2020) instead of a decoder[2]. While contrastive methods have already been applied to temporal series (van den Oord et al., 2018; Opolka et al., 2019; Gao & Shardt, 2022; Yang et al., 2023), our work additionally provides a formal characterization of InfoMax representations of Markov processes.

Another key contribution of our work lies in the introduction of an explicit bottleneck term to remove superfluous fast features. The proposed T-IB approach builds upon Wang et al. (2019b), which first proposes a reconstruction-based information bottleneck objective for molecular time series, utilizing a dimensionality-reducing linear encoder instead of a flexible deep neural architecture to implicitly reduce information. Wang & Tiwary (2021) later developed a related bottleneck objective, focusing on future target reconstruction instead of autoinformation maximization and using a marginal prior for compression. Although less reliant on the decoder architecture, this objective is not guaranteed to produce accurate simulation for arbitrary targets, as demonstrated in Appendix D.3.

## 4 EXPERIMENTAL RESULTS

We perform experiments on (i) a controlled dynamical system consisting of non-linear mixing of slow and fast processes, and (ii) molecular simulations of peptides. Our goal is, primarily, to examine the effect of the information maximization strategy (linear vs. contrastive) and the impact of the bottleneck regularization on the trajectories unfolded using LS. We further aim to validate our theory by estimating autoinformation and superfluous information for the models considered in this analysis.

**Models** We analyze representations obtained using correlation maximization methods based either on linear projections (TICA) (Molgedey & Schuster, 1994) or non-linear encoders (VAMPNet) (Mardt et al., 2018) against and non-linear autoinformation maximization (T-InfoMax) and corresponding bottleneck (T-IB) based on InfoNCE. The regularization strength $\beta$ is selected based on the validation scores[3]. We use a conditional Flow++ architecture (Ho et al., 2019) to model the variational transition distribution $q_\phi(\mathbf{z}_t|\mathbf{z}_{t-\tau})$. This is because of the modeling flexibility, the tractability of the likelihood, and the possibility of directly sampling to unfold latent simulations. Multi-layer perceptrons (MLPs)

---

[2]We refer the reader to Appendix D.2 for further details.

[3]Ablation studies on the effect of $\beta$ and the effect of a stochastic encoder can be found in Appendix F.1.

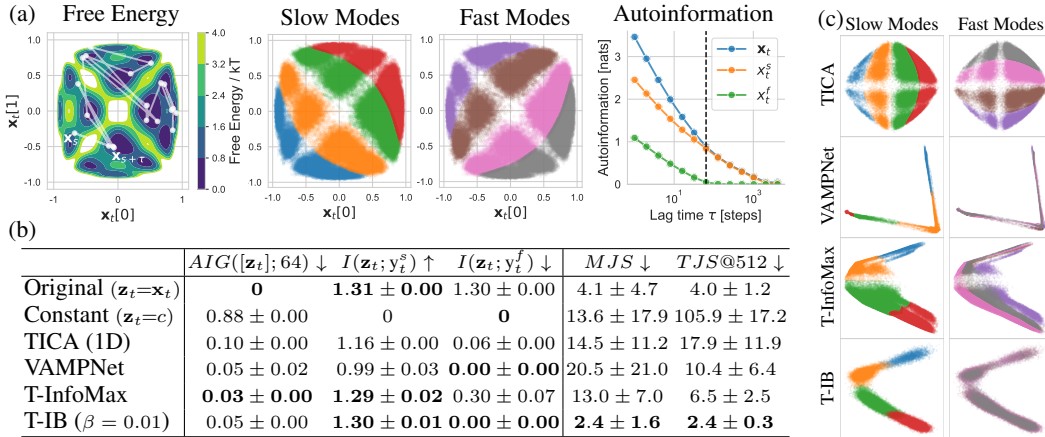

| | $AIG([\mathbf{z}_t];64)\downarrow$ | $I(\mathbf{z}_t;\mathbf{y}_t^s)\uparrow$ | $I(\mathbf{z}_t;\mathbf{y}_t^f)\downarrow$ | $MJS\downarrow$ | $TJS@512\downarrow$ |
|---|---|---|---|---|---|
| Original ($\mathbf{z}_t{=}\mathbf{x}_t$) | **0** | $\mathbf{1.31\pm0.00}$ | $1.30\pm0.00$ | $4.1\pm4.7$ | $4.0\pm1.2$ |
| Constant ($\mathbf{z}_t{=}c$) | $0.88\pm0.00$ | 0 | 0 | $13.6\pm17.9$ | $105.9\pm17.2$ |
| TICA (1D) | $0.10\pm0.00$ | $1.16\pm0.00$ | $0.06\pm0.00$ | $14.5\pm11.2$ | $17.9\pm11.9$ |
| VAMPNet | $0.05\pm0.02$ | $0.99\pm0.03$ | $\mathbf{0.00\pm0.00}$ | $20.5\pm21.0$ | $10.4\pm6.4$ |
| T-InfoMax | $\mathbf{0.03\pm0.00}$ | $1.29\pm0.02$ | $0.30\pm0.07$ | $13.0\pm7.0$ | $6.5\pm2.5$ |
| T-IB ($\beta=0.01$) | $0.05\pm0.00$ | $\mathbf{1.30\pm0.01}$ | $\mathbf{0.00\pm0.00}$ | $\mathbf{2.4\pm1.6}$ | $\mathbf{2.4\pm0.3}$ |

Figure 3: Visualization of the results on the Prinz 2D dataset. 3a: free energy and short sample trajectory (left), samples colored by the slow and fast mode index (center), and autoinformation for the full process and its components at several lag times (right). 3b: measures of autoinformation gap, mutual information between the representation and the discrete fast and slow modes in nats, and value of marginal and transition $JS$ divergence for unfolded sequences in milli-nats. 3c: trajectories encoded in the latent space $\mathbf{z}_t$ through various trained models. Quantitative and qualitative results confirm that T-IB uniquely captures relevant (slow) information while discarding irrelevant (fast) components. This results in more accurate LS as measured by the marginal and transition $JS$.

are used to model $q_\psi(\mathbf{y}_t|\mathbf{z}_t)$, mapping the representations $\mathbf{z}_t$ into the logits of a categorical distribution over the target $\mathbf{y}_t$. For all objectives, we use the same encoder, transition, and predictive architectures.

**Training** We first train the parameters $\theta$ of the encoder $p_\theta(\mathbf{z}_t|\mathbf{x}_t)$ using each objective until convergence. Note that T-IB also optimizes the parameters of the transition model $q_\phi(\mathbf{z}_t|\mathbf{z}_{t-\tau})$ during this step (as shown in equation 9). Secondly, we fix the parameters $\theta$ and fit the variational transition $q_\phi(\mathbf{z}_t|\mathbf{z}_{t-\tau})$ and predictive $q_\psi(\mathbf{y}_t|\mathbf{z}_t)$ distributions. This second phase is identical across all the models, which are trained until convergence within a maximum computational budget (50 epochs) with the AdamW optimizer (Loshchilov & Hutter, 2019) and early stopping based on the validation score. Standard deviations are obtained by running 3 experiments for each tested configuration with different seeds. Additional details on architectures and optimization can be found in Appendix E.2.

**Quantitative evaluation** We estimate the autoinformation of the representations $AI([\mathbf{z}_t]_{t=s}^T;\tau)$ at several lag time $\tau$ using SMILE (Song & Ermon, 2020) and measure the amount of information that the representations contain about the targets of interest $I(\mathbf{z}_t;\mathbf{y}_t)$ using difference of discrete entropies: $H(\mathbf{y}_t)-H(\mathbf{y}_t|\mathbf{z}_t)$ (Poole et al., 2019; McAllester & Stratos, 2020). Given an initial system state $\mathbf{x}_s$ of a test trajectory $[\mathbf{x}_t]_{t=s}^T$ and the sequence of corresponding targets $[y_t]_{t=s}^T$, we use the trained encoder, transition, and prediction models to unfold trajectories $[\tilde{y}_{s+k\tau}]_{k=1}^K \sim q^{LS}([\mathbf{y}_{s+k\tau}]_{k=1}^K|\mathbf{x}_s)$ that cover the same temporal span as the test trajectory ($K=\lfloor(T-s)/\tau\rfloor$). Similarly to previous work (Arts et al., 2023), for evaluation purposes, we consider only discrete targets $\mathbf{y}_t$ so that we can estimate the marginal and transition probabilities for the ground truth and unfolded target trajectories by counting the frequency of each target state and the corresponding transition matrix (Figure 5a). We evaluate the fidelity of the unfolded simulation by considering the Jensen-Shannon divergence ($JS$) between the ground truth and unfolded target marginal ($MJS$) and target transition distribution for several $\tau'>\tau$ ($TJS@\tau'$). Further details on the evaluation procedures are reported in Appendix E.3.

**2D Prinz Potential** Inspired by previous work (Mardt et al., 2018; Wu et al., 2018) we design a 2D process consisting of a fast $\mathbf{x}_t^f$ and slow $\mathbf{x}_t^s$ components obtained from 2 independent simulations on the 1D Prinz potential (Prinz et al., 2011). This potential energy function consists of four interconnected low-energy regions, which serve as the discrete targets $\mathbf{y}_t^f$ and $\mathbf{y}_t^s$. The two components are mixed through a linear projection and a $tanh$ non-linearity to produce a 2D process consisting of a total of 4 (fast) $\times$ 4 (slow) modes, visualized in Figure 3a. We generated separate training, validation, and test trajectories of 100K steps each. The encoders $p_\theta(\mathbf{z}_t|\mathbf{x}_t)$ consist of simple MLPs and $\mathbf{z}_t$ is fixed

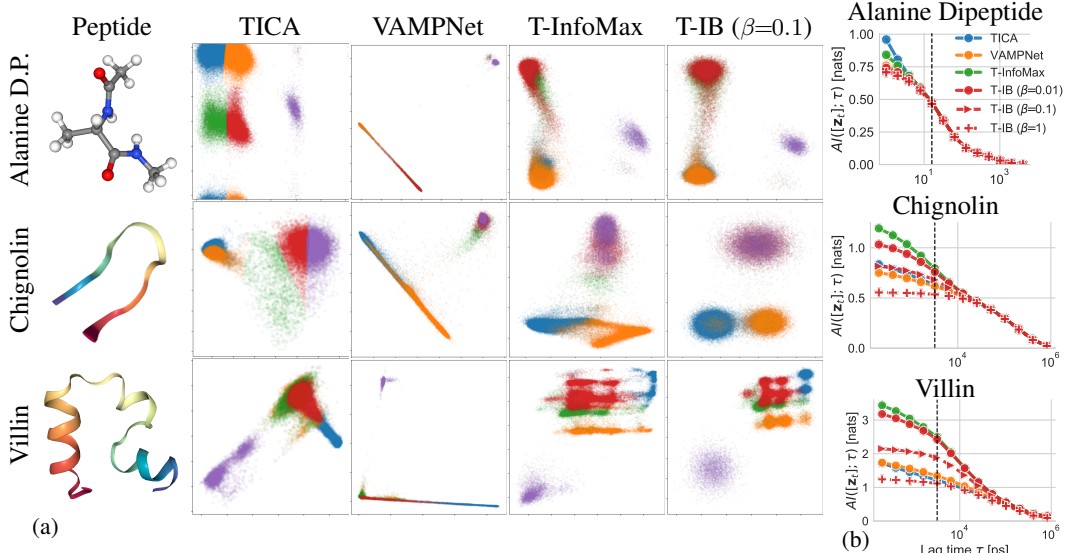

Figure 4: Comparison of 2D representations for Alanine Dipeptide, Chignolin, and Villin simulations. 4a: visualizations are colored by molecular configuration clusters $\boldsymbol{y}_t$ obtained by clustering torsion angles (Alanine Dipeptide) and TICA projections (Chignolin, Villin). 4b: corresponding values of autoinformation (y-axis) at multiple lag times (x-axis). An optimal representation should maximize autoinformation at the trained lag time $\tau$ (indicated by the dashed vertical line) while minimizing information on faster processes (to the left of the dashed line). Correlation maximization methods struggle to capture all relevant dynamics in larger systems, while T-IB regularization can effectively regulate the amount fast information in $\mathbf{z}_t$. Visually this results in simpler clustered regions.

to be 2D. As shown in the autoinformation plot in Figure 3a (on the right), at the choesen train lag time ($\tau = 64$, vertical dashed line), the fast components are temporally independent, and all the relevant information is given by the slow process: $AI(\mathbf{x}_t; 64) \approx AI(\mathbf{x}_t^s; 64) > AI(\mathbf{x}_t^f; 64) \approx 0$. Therefore, information regarding $\mathbf{x}_t^f$ can be considered superfluous (equation 7), and should be discarded.

Figure 3c visualizes the representations obtained with several models colored by the slow (left column) and fast (right column) mode index $\mathbf{y}_t^s$ and $\mathbf{y}_t^f$. We can visually observe that our proposed T-IB model preserves information regarding the slow process while removing all information regarding the irrelevant faster component. This is quantitatively supported by the measurements of mutual information reported in Table 3b, which also reports the values of marginal and transition $JS$ divergence for the unfolded slow targets trajectories $[\tilde{y}_t^s]_{t=s}^T$. We observe that the latent simulations unfolded from the T-IB representations are statistically more accurate, improving even upon trajectories unfolded by fitting the transition distribution directly in the original space $\mathbf{x}_t$. We believe this improvement is due to the substantial simplification caused by the T-IB regularization.

**Molecular Simulations** We analyze trajectories obtained by simulating *Alanine Dipeptide* and two fast-folding mini-proteins, namely *Chignolin* and *Villin* (Lindorff-Larsen et al., 2011) in water solvent. We define disjoint *train*, *validation*, and *test* splits for each molecule by splitting trajectories into temporally distinct regions. Encoders $p_\theta(\mathbf{z}_t|\mathbf{x}_t)$ employ a TorchMD Equivariant Transformer architecture (Thölke & Fabritiis, 2022) for rotation, translation, and reflection invariance. Following previous work (Köhler et al., 2023), TICA representations are obtained by projecting invariant features such as inter-atomic distances and torsion angles. Following Arts et al. (2023), the targets $\mathbf{y}_t$ are created by clustering 32-dimensional TICA projections using K-means with 5 centroids. Further details on the data splits, features and targets can be found in Appendix E.1.2.

In Figure 4, we show 2D representations obtained by training the encoders on the molecular trajectories (Figure 4a), and the corresponding measure of autoinformation (Figure 4b) at several time scales (x-axis), while Figure 5 reports transition and marginal $JS$ for trajectories unfolded on larger latent spaces (16D for Alanine simulations and 32D for Chignolin and Villin). While previous work demon-

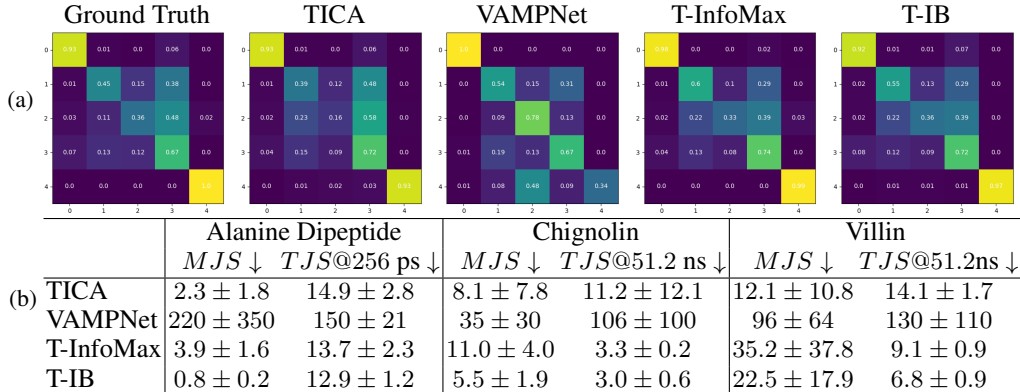

Figure 5: Evaluation of the statistical fidelity of unfolded molecular trajectories. 5a: visualization of transition matrices for ground-truth and VLS target trajectories for different models on Villin at 51.2 ns. 5b: corresponding values of marginal and transition $JS$ on Alanine Dipeptide, Chignolin and Villin. LS based on T-IB representations consistently results in lower simulation error, improving upon linear methods and unregularized T-InfoMax models.

strated that a linear operator can theoretically approximate expected system dynamics on large latent spaces (Koopman, 1931; Mezić, 2005), we note that models trained to maximize linear correlation (TICA, VAMPNet) face difficulties in extracting dynamic information in low dimensions even with non-linear encoders. Moreover, our empirical observations indicate that higher-dimensional representations obtained with VAMPNet yield transition and prediction distributions that are more difficult to fit (see Table 5 and Appendix F) resulting in less accurate unfolded target trajectories. Methods based on non-linear contrastive T-InfoMax produce more expressive representations in low dimensions. The addition of a bottleneck term aids in regulating the amount of information on faster processes (Figure 4b, left of the dashed line). As shown in Figure 5a and Table 5b, T-IB consistently improves the transition and marginal statistical accuracy when compared to the unregularized T-InfoMax counterpart. Results for additional targets and train lag times are reported in Appendix F. We estimated that training and unfolding Villin latent simulations of the same length of the training trajectory with T-IB take approximately 6 hours on a single GPU. In contrast, running molecular dynamics on the same hardware takes about 2-3 months. Further details on the run times can be found in Appendix G.

## 5 CONCLUSIONS

In this work, we propose an inference scheme designed to accelerate the simulation of Markov processes by mapping observations into a representation space where larger time steps can be modeled directly. We explore the problem of creating such a representation from an information-theoretic perspective, defining a novel objective aimed at preserving relevant dynamics while limiting superfluous information content through an Information Bottleneck. We demonstrate the effectiveness of our method from both representation learning and latent inference perspectives by comparing the information content and statistics of unfolded trajectories on synthetic data and molecular dynamics.

**Limitations and Future work** The primary focus of this work is characterizing and evaluating the dynamic properties of representations. Nevertheless, modeling accurate transition in the latent space remains a crucial aspect, and we believe that more flexible classes of transition models could result in higher statistical fidelity at the cost of slower sampling. Another challenging aspect involves creating representations of systems with large autoinformation content (e.g. chaotic and unstable systems). This is because the variance of modern mutual information lower bounds increases exponentially with the amount of information to extract (McAllester & Stratos, 2020). To mitigate this issue and validate the applicability of our method to other practical settings, future work will consider exploiting local similarity and studying the generalization capabilities of models trained on multiple systems and different simulation conditions. We further aim to evaluate the accuracy of long-range unfolded trajectories when only collections of shorter simulations are available during training time.

ACKNOWLEDGMENTS

We thank Frank Noé, Rianne van den Berg, Victor Garcia Satorras, Chin-Wei Huang, Marloes Arts, Wessel Bruinsma, Tian Xie, and Claudio Zeni for the insightful discussions and feedback provided throughout the project. This work was supported by the Microsoft Research PhD Scholarship Programme.

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

## A    NOTATION

Throughout the paper we use the following notation to indicate mutual information:

$$I(\mathbf{x}; \mathbf{z}) := \mathrm{KL}(p(\mathbf{x}, \mathbf{z}) || p(\mathbf{x})p(\mathbf{z}))$$

$$= \mathbb{E}_{p(\mathbf{x}, \mathbf{z})} \left[ \log \frac{p(\mathbf{x}, \mathbf{z})}{p(\mathbf{x})p(\mathbf{z})} \right]. \tag{10}$$

To improve readability we omit the subscript for the expectation. Unless otherwise specified, expectations are computed with respect to the ground-truth distribution $p(\mathbf{x}, \mathbf{z})$.

Similarly, we leave the expectation for conditional KL-divergence implicit:

$$\mathrm{KL}(p(\mathbf{x}|\mathbf{z}) || q(\mathbf{x}|\mathbf{z})) := \mathbb{E}_{p(\mathbf{x}, \mathbf{z})} \left[ \log \frac{p(\mathbf{x}|\mathbf{z})}{q(\mathbf{x}|\mathbf{z})} \right]. \tag{11}$$

We will use $\epsilon$ to indicate a stochastic source external to the system, i.e. $I(\epsilon; \cdot) = 0$ with $\cdot$ as a placeholder for any variable (or combination of variables) in the system (excluding $\epsilon$ itself).

## B    PROOFS

We start by introducing the assumptions and properties that are used throughout the section. Then, we list proofs for the statements in the main text.

### B.1    GENERAL ASSUMPTIONS

As a preliminary step for proving the statements in Section 2, we clarify our general assumptions

**(A.1)** $\mathbf{z}_t$ is a representation of $\mathbf{x}_t$.
With this statement, we signify that $\mathbf{z}_t$ can be expressed as a noisy function of $\mathbf{x}_t$: $\mathbf{z}_t = f(\mathbf{x}_t, \epsilon_t)$. This implies that $\mathbf{z}_t$ is conditionally independent of any other variable of the system when $\mathbf{x}_t$ is observed: $I(\mathbf{z}_t; \cdot | \mathbf{x}_t, \cdot) = 0$, in which $\cdot$ is a placeholder for other variables (or combinations thereof) in the system.

**(A.2)** $\mathbf{y}_t$ is a representation of $\mathbf{x}_t$.
Analogously to the previous assumption, we assume that the target of interest can be expressed as a noisy function of $\mathbf{x}_t$. Therefore we will assume the same corresponding conditional independence.

**(A.3)** $[\mathbf{x}_t]_{t=s}^{T}$ form a homogeneous Markov Chain.
This assumption can be expressed in terms of conditional independence between past events $[\mathbf{x}_t]_{t=s}^{m-1}$, and future $\mathbf{x}_{m+1}$ when the current event $\mathbf{x}_m$ is observed:

$$I([\mathbf{x}_t]_{t=s}^{m-1}; \mathbf{x}_{m+1} | \mathbf{x}_m) = 0,$$

for any $m \in [s+1, T-1]$.

### B.2    PROPERTIES

For completeness, here we list properties of mutual information that are used to prove the statements in the following sections. Let $\mathbf{a}, \mathbf{b}, \mathbf{c}, \mathbf{d}$ be random variables with some joint distribution $p(\mathbf{a}, \mathbf{b}, \mathbf{c}, \mathbf{d})$.

**(P.1)** Non-negativity of (conditional) mutual information.
Mutual information (and conditional mutual information) is non-negative:

$$I(\mathbf{a}; \mathbf{b}|\mathbf{c}) \geq 0 \tag{12}$$

**(P.2)** Chain rule of mutual information.
Mutual information (and conditional mutual information) can be factorized as follows:

$$I(\mathbf{ab}; \mathbf{c}|\mathbf{d}) = I(\mathbf{a}; \mathbf{c}|\mathbf{d}) + I(\mathbf{b}; \mathbf{c}|\mathbf{ad})$$

$$= I(\mathbf{b}; \mathbf{c}|\mathbf{d}) + I(\mathbf{a}; \mathbf{c}|\mathbf{bd}) \tag{13}$$

**(P.3)** Data processing inequality (DPI).

Mutual information (and conditional mutual information) between two random variables cannot increase by applying functions to either argument (on the left side of the conditioning). In this paper, we will use a slightly more general version of DPI in which we also consider noisy functions (with independent noise):

$$I(\mathbf{a}; \mathbf{b}|\mathbf{c}) \geq I(f(\mathbf{a}, \boldsymbol{\epsilon}); \mathbf{b}|\mathbf{c}) \tag{14}$$

### B.3 Autoinformation and Data Processing Inequality

Here we demonstrate that the autoinformation in the original space $AI(\mathbf{x}_t; \tau)$ is an upper bound for the autoinformation of the representation $AI(\mathbf{z}_t; \tau)$.

**Statement.** *The autoinformation for $\mathbf{x}_t$ upper-bounds the autoinformation for the corresponding representation $\mathbf{z}_t$*

$$AI(\mathbf{x}_t; \tau) \geq AI(\mathbf{z}_t; \tau) \tag{15}$$

*Proof.*

$$
\begin{aligned}
AI(\mathbf{x}_t; \tau) &= I(\mathbf{x}_t; \mathbf{x}_{t+\tau}) \\
&\overset{(P.2)}{=} I(\mathbf{x}_t \mathbf{z}_t; \mathbf{x}_{t+\tau}) - I(\mathbf{z}_t; \mathbf{x}_{t+\tau}|\mathbf{x}_t) \\
&\overset{(A.1)}{=} I(\mathbf{x}_t \mathbf{z}_t; \mathbf{x}_{t+\tau}) \\
&\overset{(P.2)}{=} I(\mathbf{z}_t; \mathbf{x}_{t+\tau}) + I(\mathbf{x}_t; \mathbf{x}_{t+\tau}|\mathbf{z}_t) \\
&\overset{(P.2)}{=} I(\mathbf{z}_t; \mathbf{x}_{t+\tau} \mathbf{z}_{t+\tau}) - I(\mathbf{z}_t; \mathbf{z}_{t+\tau}|\mathbf{x}_{t+\tau}) + I(\mathbf{x}_t; \mathbf{x}_{t+\tau}|\mathbf{z}_t) \\
&\overset{(A.1)}{=} I(\mathbf{z}_t; \mathbf{x}_{t+\tau} \mathbf{z}_{t+\tau}) + I(\mathbf{x}_t; \mathbf{x}_{t+\tau}|\mathbf{z}_t) \\
&\overset{(P.2)}{=} I(\mathbf{z}_t; \mathbf{z}_{t+\tau}) + I(\mathbf{z}_t; \mathbf{x}_{t+\tau}|\mathbf{z}_{t+\tau}) + I(\mathbf{x}_t; \mathbf{x}_{t+\tau}|\mathbf{z}_t) \\
&= AI(\mathbf{z}_t; \tau) + I(\mathbf{z}_t; \mathbf{x}_{t+\tau}|\mathbf{z}_{t+\tau}) + I(\mathbf{x}_t; \mathbf{x}_{t+\tau}|\mathbf{z}_t).
\end{aligned} \tag{16}
$$

Using property (P.1), we infer $AI(\mathbf{x}_t; \tau) \geq AI(\mathbf{z}_t; \tau)$ □

**Remark 4.** *The autoinformation gap upper bounds the amount of information that $\mathbf{x}_t$ and $\mathbf{x}_{t+\tau}$ share whenever one of the two corresponding representations is observed:*

$$AIG(\mathbf{z}_t; \tau) \geq I(\mathbf{x}_t; \mathbf{x}_{t+\tau}|\mathbf{z}_t), \tag{17}$$

*and*

$$AIG(\mathbf{z}_t; \tau) \geq I(\mathbf{x}_t; \mathbf{x}_{t+\tau}|\mathbf{z}_{t+\tau}) \tag{18}$$

*Proof.* Re-arranging the terms in equation 16 we can also characterize the autoinformation gap as:

$$
\begin{aligned}
AI(\mathbf{x}_t; \tau) - AI(\mathbf{z}_t; \tau) &= I(\mathbf{z}_t; \mathbf{x}_{t+\tau}|\mathbf{z}_{t+\tau}) + I(\mathbf{x}_t; \mathbf{x}_{t+\tau}|\mathbf{z}_t) \tag{19} \\
&= I(\mathbf{x}_t; \mathbf{z}_{t+\tau}|\mathbf{z}_t) + I(\mathbf{x}_t; \mathbf{x}_{t+\tau}|\mathbf{z}_{t+\tau}). \tag{20}
\end{aligned}
$$

The expression in the second line can be derived by symmetry. Statement 4 follows from (P.1). □

### B.4 Autoinformation of sequences

**Statement.** *A sequence of representations preserves autoinformation at $\tau$ if and only if all the pairs of its elements at temporal distance $\tau$ preserve autoinformation*

$$AIG([\mathbf{z}_t]_{t=s}^{T}; \tau) = 0 \iff AIG(\mathbf{z}_m; \tau) = 0 \quad \forall m \in [s, T - \tau] \tag{21}$$

*Proof.* We prove the two directions of the implication separately:

- $\implies$

  Proof by contradiction. Assume

**(T.1)** $\exists m \in [s, T - \tau]$ for which the autoinformation in $\mathbf{x}_m$ is strictly larger than the autoinformation of the corresponding representation $\mathbf{z}_m$

$$AI(\mathbf{x}_m; \tau) > AI(\mathbf{z}_m; \tau)$$

.

The autoinformation gap between the two sequences can be written as:

$$
\begin{aligned}
AIG([\mathbf{z}_t]_{t=s}^{T}; \tau) &= \mathbb{E}_t \left[ AI(\mathbf{x}_t; \tau) - AI(\mathbf{z}_t; \tau) \right] \\
&\overset{B.3}{\geq} \frac{1}{T - s - \tau + 1} \left( AI(\mathbf{x}_m; \tau) - AI(\mathbf{z}_m; \tau) \right) \\
&\overset{(T.1)}{>} 0.
\end{aligned}
\tag{22}
$$

We derived that the autoinformation gap must be strictly positive, which results in a contradiction.

- $\impliedby$
  If we assume that mutual information is the same for all the pairs, clearly their average is also the same.

$\square$

## B.5    AUTOINFORMATION AND SUFFICIENCY

**Statement.** *Whenever $\mathbf{z}_t$ preserves autoinformation, $\mathbf{z}_t$ is sufficient for any (noisy) function of $\mathbf{x}_{t+\tau}$:*

$$AIG(\mathbf{z}_t; \tau) \iff I(\mathbf{x}_t; \mathbf{y}_{t+\tau}) = I(\mathbf{z}_t; \mathbf{y}_{t+\tau}) \quad \forall \mathbf{y}_{t+\tau} := f(\mathbf{x}_{t+\tau}, \epsilon).$$

*Proof.* We address the two directions of the implication in Lemma 1 separately:

- $\implies$
  We start by assuming that $\mathbf{z}_t$ preserves information at $\tau$: $AI(\mathbf{x}_t; \tau) - AI(\mathbf{z}_t; \tau) = 0$

$$
\begin{aligned}
I(\mathbf{x}_t; \mathbf{y}_{t+\tau}) &\overset{(P.2)}{=} I(\mathbf{x}_t \mathbf{z}_t; \mathbf{y}_{t+\tau}) - I(\mathbf{z}_t; \mathbf{y}_{t+\tau} | \mathbf{x}_t) \\
&\overset{(P.1)}{\leq} I(\mathbf{x}_t \mathbf{z}_t; \mathbf{y}_{t+\tau}) \\
&\overset{(P.2)}{=} I(\mathbf{z}_t; \mathbf{y}_{t+\tau}) + I(\mathbf{x}_t; \mathbf{y}_{t+\tau} | \mathbf{z}_t) \\
&\overset{(P.3)}{\leq} I(\mathbf{z}_t; \mathbf{y}_{t+\tau}) + I(\mathbf{x}_t; \mathbf{x}_{t+\tau} | \mathbf{z}_t) \\
&\overset{4}{=} I(\mathbf{z}_t; \mathbf{y}_{t+\tau}).
\end{aligned}
$$

  Since we showed $I(\mathbf{x}_t; \mathbf{y}_{t+\tau}) \leq I(\mathbf{z}_t; \mathbf{y}_{t+\tau})$, and using DPI we have $I(\mathbf{x}_t; \mathbf{y}_{t+\tau}) \geq I(\mathbf{z}_t; \mathbf{y}_{t+\tau})$, we must conclude that $I(\mathbf{x}_t; \mathbf{y}_{t+\tau}) = I(\mathbf{z}_t; \mathbf{y}_{t+\tau})$

- $\impliedby$ We prove the second direction of the double implication by contradiction.

**(T.1)** Let $\mathbf{y}_{t+\tau}$ be a noisy function of $\mathbf{x}_{t+\tau}$ for which $\mathbf{z}_t$ is not sufficient:

$$I(\mathbf{x}_t; \mathbf{y}_{t+\tau}) > I(\mathbf{z}_t; \mathbf{y}_{t+\tau})$$

then

$$AI(\mathbf{z}_t; \tau) = I(\mathbf{z}_t; \mathbf{z}_{t+\tau})$$

$$\overset{(P.2)}{=} I(\mathbf{z}_t; \mathbf{y}_{t+\tau}\mathbf{z}_{t+\tau}) - I(\mathbf{z}_t; \mathbf{y}_{t+\tau}|\mathbf{z}_{t+\tau})$$

$$\overset{(P.1)}{\leq} I(\mathbf{z}_t; \mathbf{y}_{t+\tau}\mathbf{z}_{t+\tau})$$

$$\overset{(P.2)}{=} I(\mathbf{z}_t; \mathbf{y}_{t+\tau}) + I(\mathbf{z}_t; \mathbf{z}_{t+\tau}|\mathbf{y}_{t+\tau})$$

$$\overset{(T.1)}{<} I(\mathbf{x}_t; \mathbf{y}_{t+\tau}) + I(\mathbf{z}_t; \mathbf{z}_{t+\tau}|\mathbf{y}_{t+\tau})$$

$$\overset{(P.3)}{\leq} I(\mathbf{x}_t; \mathbf{y}_{t+\tau}) + I(\mathbf{x}_t; \mathbf{x}_{t+\tau}|\mathbf{y}_{t+\tau})$$

$$\overset{(P.2)}{=} I(\mathbf{x}_t; \mathbf{x}_{t+\tau}\mathbf{y}_{t+\tau})$$

$$\overset{(P.2)}{=} I(\mathbf{x}_t; \mathbf{x}_{t+\tau}) + I(\mathbf{x}_t; \mathbf{y}_{t+\tau}|\mathbf{x}_{t+\tau})$$

$$\overset{(A.2)}{=} AI(\mathbf{x}_t; \tau). \tag{23}$$

We derived that the $AIG(\mathbf{z}_t; \tau) > 0$, which contradicts the premises, concluding the proof.

$\square$

## B.6 MARKOV PROPERTY

**Statement.** *Sequences of representations of a homogeneous Markov Chain that preserve information at some lag time $\tau$ also form a homogeneous Markov Chain at temporal resolution $\tau$:*

$$AIG([\mathbf{z}_t]_{t=s}^{T}; \tau) = 0 \implies [\mathbf{z}_{s'+k\tau}]_{k=0}^{K} \text{ is a homogeneous Markov Chain,} \tag{24}$$

*with $s' \in [s, T - \tau]$, $K \leq \lfloor (T - s')/\tau \rfloor$.*

*Proof.* In order to prove that $[\mathbf{z}_{s'+k\tau}]_{k=0}^{K}$ form a homogeneous Markov Chain, we first show that $[\mathbf{z}_{s'+k\tau}]_{k=0}^{K}$ satisfies the Markov property. This can be shown by upper-bounding the amount of information that the past $[\mathbf{z}_{s'+j\tau}]_{j=0}^{k-1}$ carries about the next representation $\mathbf{z}_{s'+(k+1)\tau}$ whenever the current representation $\mathbf{z}_{s'+k\tau}$ is observed:

$$I\big([\mathbf{z}_{s'+j\tau}]_{j=0}^{k-1}; \mathbf{z}_{s'+(k+1)\tau}|\mathbf{z}_{s'+k\tau}\big) \overset{(P.3)}{\leq} I\big([\mathbf{x}_{s'+j\tau}]_{j=0}^{k-1}; \mathbf{x}_{s'+(k+1)\tau}|\mathbf{z}_{s'+k\tau}\big)$$

$$\overset{(P.2)}{=} I\big([\mathbf{x}_{s'+j\tau}]_{j=0}^{k-1}\mathbf{z}_{s'+k\tau}; \mathbf{x}_{s'+(k+1)\tau}\big) - I\big(\mathbf{z}_{s'+k\tau}; \mathbf{x}_{s'+(k+1)\tau}\big)$$

$$\overset{(P.3)}{\leq} I\big([\mathbf{x}_{s'+j\tau}]_{j=0}^{k}; \mathbf{x}_{s'+(k+1)\tau}\big) - I\big(\mathbf{z}_{s'+k\tau}; \mathbf{x}_{s'+(k+1)\tau}\big)$$

$$\overset{(P.2)}{=} I\big(\mathbf{x}_{s'+k\tau}; \mathbf{x}_{s'+(k+1)\tau}\big) - I\big(\mathbf{z}_{s'+k\tau}; \mathbf{x}_{s'+(k+1)\tau}\big)$$

$$+ I\big([\mathbf{x}_{s'+j\tau}]_{j=0}^{k-1}; \mathbf{x}_{s'+(k+1)\tau}|\mathbf{x}_{s'+k\tau}\big)$$

$$\overset{(A.3)}{=} I\big(\mathbf{x}_{s'+k\tau}; \mathbf{x}_{s'+(k+1)\tau}\big) - I\big(\mathbf{z}_{s'+k\tau}; \mathbf{x}_{s'+(k+1)\tau}\big)$$

$$\overset{(P.3)}{\leq} I\big(\mathbf{x}_{s'+k\tau}; \mathbf{x}_{s'+(k+1)\tau}\big) - I\big(\mathbf{z}_{s'+k\tau}; \mathbf{z}_{s'+(k+1)\tau}\big)$$

$$= AI(\mathbf{x}_{s'+k\tau}; \tau) - AI(\mathbf{z}_{s'+k\tau}; \tau)$$

$$= AIG(\mathbf{z}_{s'+k\tau}; \tau) = 0, \tag{25}$$

for any $s' \in [s, T - \tau]$, $K \leq \lfloor (T - s')/\tau \rfloor$, and $k \in [1, K - 1]$.

Using the results from B.4 and the premise that the autoinformation gap is zero, we can conclude that the conditional mutual information in the previous equation must be zero, and the Markov property holds. Furthermore since both $p(\mathbf{x}_t|\mathbf{x}_{t-\tau})$ and $p(\mathbf{z}_t|\mathbf{x}_t)$ are time-independent, we must conclude that $p(\mathbf{z}_t|\mathbf{z}_{t-\tau})$ must satisfy the same property. Therefore, we conclude that $[\mathbf{z}_{s'+k\tau}]_{k=0}^{K}$ forms a homogeneous Markov Chain. $\square$

## B.7 BOUNDS ON THE AUTOINFORMATION GAP

**Statement.** *For any $\tau' > \tau > 0$, the autoinformation gap for $\mathbf{z}_t$ at lag time $\tau'$ is upper-bounded by the sum of the autoinformation gap for $\mathbf{z}_t$ at lag time $\tau$ and the autoinformation gap for $\mathbf{z}_{t+\tau-\tau}$ at lag time $\tau$:*

$$AIG(\mathbf{z}_t; \tau') \leq AIG(\mathbf{z}_t; \tau) + AIG(\mathbf{z}_{t+\tau'-\tau}; \tau) \tag{26}$$

*Proof.* Let $\tau' > \tau > 0$. The autoinformation for $\mathbf{x}_t$ at $\tau'$ can be written as:

$$
\begin{aligned}
AI(\mathbf{x}_t; \tau') &= I(\mathbf{x}_t; \mathbf{x}_{t+\tau'}) \\
&\overset{(P.2)}{=} I(\mathbf{x}_t \mathbf{z}_t; \mathbf{x}_{t+\tau'}) - I(\mathbf{z}_t; \mathbf{x}_{t+\tau'} | \mathbf{x}_t) \\
&\overset{(P.1)}{\leq} I(\mathbf{x}_t \mathbf{z}_t; \mathbf{x}_{t+\tau'}) \\
&\overset{(P.2)}{=} I(\mathbf{z}_t; \mathbf{x}_{t+\tau'}) + I(\mathbf{x}_t; \mathbf{x}_{t+\tau'} | \mathbf{z}_t) \\
&\overset{(P.3)}{\leq} I(\mathbf{z}_t; \mathbf{x}_{t+\tau'}) + I(\mathbf{x}_t; \mathbf{x}_{t+\tau} | \mathbf{z}_t) \\
&\overset{4}{\leq} I(\mathbf{z}_t; \mathbf{x}_{t+\tau'}) + AIG(\mathbf{z}_t; \tau) \\
&\overset{(P.2)}{=} I(\mathbf{z}_t; \mathbf{x}_{t+\tau'} \mathbf{z}_{t+\tau'}) - I(\mathbf{z}_t; \mathbf{z}_{t+\tau'} | \mathbf{x}_{t+\tau'}) + AIG(\mathbf{z}_t; \tau) \\
&\overset{(P.1)}{\leq} I(\mathbf{z}_t; \mathbf{x}_{t+\tau'} \mathbf{z}_{t+\tau'}) + AIG(\mathbf{z}_t; \tau) \\
&\overset{(P.2)}{=} I(\mathbf{z}_t; \mathbf{z}_{t+\tau'}) + I(\mathbf{z}_t; \mathbf{x}_{t+\tau'} | \mathbf{z}_{t+\tau'}) + AIG(\mathbf{z}_t; \tau) \\
&\overset{(P.3)}{\leq} I(\mathbf{z}_t; \mathbf{z}_{t+\tau'}) + I(\mathbf{x}_t; \mathbf{x}_{t+\tau'} | \mathbf{z}_{t+\tau'}) + AIG(\mathbf{z}_t; \tau) \\
&\overset{(P.3)}{\leq} I(\mathbf{z}_t; \mathbf{z}_{t+\tau'}) + I(\mathbf{x}_{t+\tau'-\tau}; \mathbf{x}_{t+\tau'} | \mathbf{z}_{t+\tau'}) + AIG(\mathbf{z}_t; \tau) \\
&\overset{4}{\leq} AI(\mathbf{z}_t; \tau') + AIG(\mathbf{z}_t; \tau) + AIG(\mathbf{z}_{t+\tau'-\tau}; \tau).
\end{aligned}
\tag{27}
$$

Re-arranging the terms, we have:

$$AIG(\mathbf{z}_t; \tau') \leq AIG(\mathbf{z}_t; \tau) + AIG(\mathbf{z}_{t+\tau'-\tau}; \tau). \tag{28}$$

Note that whenever $\mathbf{z}_t$ is sampled from the equilibrium, we have $AI(\mathbf{z}_t; \tau') \leq 2AI(\mathbf{z}_t; \tau)$. $\qquad\square$

## B.8 SLOWER INFORMATION PRESERVATION

**Statement.** *If a sequence of representation preserves autoinformation at lag time $\tau$, then it preserves autoinformation for any $\tau' \geq \tau$:*

$$AIG([\mathbf{z}_t]_{t=s}^{T}; \tau) = 0 \implies AIG([\mathbf{z}_t]_{t=s}^{T}; \tau') = 0 \tag{29}$$

*Proof.* Using the result from B.7, we can express the autoinformation gap at $\tau'$ as:

$$
\begin{aligned}
AIG([\mathbf{z}_t]_{t=s}^{T}; \tau') &= \mathbb{E}_{t \sim U(s, T-\tau')}[AIG(\mathbf{z}_t; \tau')] \\
&\overset{B.7}{\leq} \mathbb{E}_{t \sim U(s, T-\tau')}[AIG(\mathbf{z}_t; \tau) + AIG(\mathbf{z}_{t+\tau'-\tau}; \tau)] \\
&= AIG([\mathbf{z}_t]_{t=s}^{T-\tau'+\tau}; \tau) + AIG([\mathbf{z}_t]_{t=s+\tau'-\tau}^{T}; \tau).
\end{aligned}
\tag{30}
$$

Since both $[\mathbf{z}_t]_{t=s}^{T-\tau'+\tau}$ and $[\mathbf{z}_t]_{t=s+\tau'-\tau}^{T}$ are sub-sequences of $[\mathbf{z}_t]_{t=s}^{T}$, and $AIG([\mathbf{z}_t]_{t=s}^{T}; \tau) = 0$, we can infer that the right side of equation 30 must be zero. Furthermore, since $AIG([\mathbf{z}_t]_{t=s}^{T}; \tau') \geq 0$, we must conclude $AIG([\mathbf{z}_t]_{t=s}^{T}; \tau') = 0$. $\qquad\square$

**Statement.** *The average latent simulation error introduced by unfolding $K$ steps using latent simulation starting from $\mathbf{x}_t$ with $t \sim U(s, s+\tau-1)$ is upper-bounded by $K$ times the autoinformation gap for the sequence $[\mathbf{z}_t]_{t=s}^{T}$, with $T = s + (K+1)\tau - 1$:*

$$\mathbb{E}_t \left[ \underbrace{\text{KL}(p([\mathbf{y}_{t+k\tau}]_{k=1}^{K} \,|\, \mathbf{x}_t) || p^{LS}([\mathbf{y}_{t+k\tau}]_{k=1}^{K} \,|\, \mathbf{x}_t))}_{\textit{Latent Simulation error for } K \textit{ steps of } \tau \textit{ starting from } t} \right] \leq K \underbrace{AIG([\mathbf{z}_t]_{t=s}^{T}; \tau)}_{\textit{Autoinformation gap for lag time } \tau} . \tag{31}$$

*Proof.* We start with the following bound:

$$\text{KL}(p([\mathbf{y}_{t+k\tau}]_{k=1}^{K} \,|\, \mathbf{x}_t) || p^{LS}([\mathbf{y}_{t+k\tau}]_{k=1}^{K} \,|\, \mathbf{x}_t)) \leq \text{KL}(p([\mathbf{x}_{t+k\tau}]_{k=1}^{K} \,|\, \mathbf{x}_t) || p^{LS}([\mathbf{x}_{t+k\tau}]_{k=1}^{K} \,|\, \mathbf{x}_t)), \tag{32}$$

which holds because of assumption (A.2) and the data processing inequality. Secondly, we upperbound the right-most term as a sum of autoinformation:

$$\text{KL}(p([\mathbf{x}_{t+k\tau}]_{k=1}^{K} \,|\, \mathbf{x}_t) || p^{LS}([\mathbf{x}_{t+k\tau}]_{k=1}^{K} \,|\, \mathbf{x}_t)) \leq \text{KL}(p(\mathbf{z}_t, [\mathbf{x}_{t+k\tau}, \mathbf{z}_{t+k\tau}]_{k=1}^{K} \,|\, \mathbf{x}_t) || p^{LS}(\mathbf{z}_t, [\mathbf{x}_{t+k\tau}, \mathbf{z}_{t+k\tau}]_{k=1}^{K} \,|\, \mathbf{x}_t))$$

$$= \mathbb{E} \left[ \log \frac{p_\theta(\mathbf{z}_t|\mathbf{x}_t) \prod_{k=1}^{K} p(\mathbf{x}_{t+k\tau}|\mathbf{x}_{t+(k-1)\tau}) p_\theta(\mathbf{z}_{t+k\tau}|\mathbf{x}_{t+k\tau})}{p_\theta(\mathbf{z}_t|\mathbf{x}_t) \prod_{k=1}^{K} p(\mathbf{z}_{t+k\tau}|\mathbf{z}_{t+(k-1)\tau}) p(\mathbf{x}_{t+k\tau}|\mathbf{z}_{t+k\tau})} \right]$$

$$= \sum_{k=1}^{K} \mathbb{E} \left[ \log \frac{p(\mathbf{x}_{t+k\tau}|\mathbf{x}_{t+(k-1)\tau})}{p(\mathbf{z}_{t+k\tau}|\mathbf{z}_{t+(k-1)\tau})} \frac{p_\theta(\mathbf{z}_{t+k\tau}|\mathbf{x}_{t+k\tau})}{p(\mathbf{x}_{t+k\tau}|\mathbf{z}_{t+k\tau})} \right]$$

$$= \sum_{k=1}^{K} \mathbb{E} \left[ \log \frac{p(\mathbf{x}_{t+k\tau}|\mathbf{x}_{t+(k-1)\tau})}{p(\mathbf{z}_{t+k\tau}|\mathbf{z}_{t+(k-1)\tau})} \frac{p(\mathbf{z}_{t+k\tau})}{p(\mathbf{x}_{t+k\tau})} \right]$$

$$= \sum_{k=1}^{K} \mathbb{E} \left[ \log \frac{p(\mathbf{x}_{t+k\tau}|\mathbf{x}_{t+(k-1)\tau})}{p(\mathbf{x}_{t+k\tau})} \right] + \mathbb{E} \left[ \log \frac{p(\mathbf{z}_{t+k\tau})}{p(\mathbf{z}_{t+k\tau}|\mathbf{z}_{t+(k-1)\tau})} \right]$$

$$= \sum_{k=0}^{K-\tau} AI(\mathbf{x}_{t+k\tau}; \tau) - AI(\mathbf{z}_{t+k\tau}; \tau)$$

$$= \sum_{k=0}^{K-\tau} AIG(\mathbf{z}_{t+k\tau}; \tau) \tag{33}$$

Lastly, we consider the average error when $t \sim U(s, s+\tau-1)$

$$\mathbb{E}_t \left[ \text{KL}(p([\mathbf{y}_{t+k\tau}]_{k=1}^{K} \,|\, \mathbf{x}_t) || p^{LS}([\mathbf{y}_{t+k\tau}]_{k=1}^{K} \,|\, \mathbf{x}_t)) \right] = \frac{1}{\tau} \sum_{t=s}^{s+\tau-1} \text{KL}(p([\mathbf{x}_{t+k\tau}]_{k=1}^{K} \,|\, \mathbf{x}_t) || p^{LS}([\mathbf{x}_{t+k\tau}]_{k=1}^{K} \,|\, \mathbf{x}_t))$$

$$= \frac{1}{\tau} \sum_{t=s}^{s+\tau-1} \sum_{k=0}^{K-\tau} AIG(\mathbf{z}_{t+k\tau}; \tau)$$

$$= \frac{1}{\tau} \sum_{t=s}^{s+K\tau-1} AIG(\mathbf{z}_{t+k\tau}; \tau)$$

$$= \frac{K\tau}{\tau} AIG([\mathbf{z}_t]_{t=s}^{s+(K+1)\tau-1}; \tau)$$

$$= K\, AIG([\mathbf{z}_t]_{t=s}^{T}; \tau), \tag{34}$$

with $T := s + (K+1)\tau - 1$. This concludes the proof. $\square$

Hereby, we outline the steps to obtain the expression reported in Equation 3:

$$
\begin{aligned}
\mathrm{KL}&(p([\mathbf{y}_{s+k\tau}]_{k=1}^{K}|\mathbf{x}_s)||q^{LS}([\mathbf{y}_{s+k\tau}]_{k=1}^{K}|\mathbf{x}_s)) \\
&= \mathbb{E}\left[\log \frac{p([\mathbf{y}_{s+k\tau}]_{k=1}^{K}|\mathbf{x}_s)}{p^{LS}([\mathbf{y}_{s+k\tau}]_{k=1}^{K}|\mathbf{x}_s)} \frac{p^{LS}([\mathbf{y}_{s+k\tau}]_{k=1}^{K}|\mathbf{x}_s)}{q^{LS}([\mathbf{y}_{s+k\tau}]_{k=1}^{K}|\mathbf{x}_s)}\right] \\
&= \mathrm{KL}(p([\mathbf{y}_{s+k\tau}]_{k=1}^{K}|\mathbf{x}_s)||p^{LS}([\mathbf{y}_{s+k\tau}]_{k=1}^{K}|\mathbf{x}_s)) + \mathbb{E}\left[\log \frac{p^{LS}([\mathbf{y}_{s+k\tau}]_{k=1}^{K}|\mathbf{x}_s)}{q^{LS}([\mathbf{y}_{s+k\tau}]_{k=1}^{K}|\mathbf{x}_s)}\right]. \quad (35)
\end{aligned}
$$

Focusing on the second term:

$$
\begin{aligned}
\mathbb{E}\left[\log \frac{p^{LS}([\mathbf{y}_{s+k\tau}]_{k=1}^{K}|\mathbf{x}_s)}{q^{LS}([\mathbf{y}_{s+k\tau}]_{k=1}^{K}|\mathbf{x}_s)}\right] &\leq \mathbb{E}\left[\log \frac{p^{LS}([\mathbf{y}_{s+k\tau}, \mathbf{z}_{s+k\tau}]_{k=1}^{K}|\mathbf{x}_s)}{q^{LS}([\mathbf{y}_{s+k\tau}, \mathbf{z}_{s+k\tau}]_{k=1}^{K}|\mathbf{x}_s)}\right] \\
&= \mathbb{E}\left[\log \frac{\prod_{k=1}^{K} p(\mathbf{z}_{s+k\tau}|\mathbf{z}_{s+(k-1)\tau})p(\mathbf{y}_{s+k\tau}|\mathbf{z}_{s+k\tau})}{\prod_{k=1}^{K} q_\phi(\mathbf{z}_{s+k\tau}|\mathbf{z}_{s+(k-1)\tau})q_\psi(\mathbf{y}_{s+k\tau}|\mathbf{z}_{s+k\tau})}\right] \\
&= \sum_{k=1}^{K} \mathbb{E}\left[\log \frac{p(\mathbf{z}_{s+k\tau}|\mathbf{z}_{s+(k-1)\tau})}{q_\phi(\mathbf{z}_{s+k\tau}|\mathbf{z}_{s+(k-1)\tau})}\right] + \mathbb{E}\left[\log \frac{p(\mathbf{y}_{s+k\tau}|\mathbf{z}_{s+k\tau})}{q_\psi(\mathbf{y}_{s+k\tau}|\mathbf{z}_{s+k\tau})}\right] \\
&= \sum_{k=1}^{K} \mathrm{KL}(p(\mathbf{z}_{s+k\tau}|\mathbf{z}_{s+(k-1)\tau})||q_\phi(\mathbf{z}_{s+k\tau}|\mathbf{z}_{s+(k-1)\tau})) \\
&\quad + \mathrm{KL}(p(\mathbf{y}_{s+k\tau}|\mathbf{z}_{s+k\tau})||q_\psi(\mathbf{y}_{s+k\tau}|\mathbf{z}_{s+k\tau})). \quad (36)
\end{aligned}
$$

The total amount of information that a representation $\mathbf{z}_t$ contains about the original data $\mathbf{x}_t$ can be de-composed using the chain rule of mutual information as follows:

$$
\begin{aligned}
I(\mathbf{x}_t; \mathbf{z}_t) &\stackrel{(P.2)}{=} I(\mathbf{x}_t\mathbf{z}_{t-\tau}; \mathbf{z}_t) - I(\mathbf{z}_t; \mathbf{z}_{t-\tau}|\mathbf{x}_t) \\
&\stackrel{(A.1)}{=} I(\mathbf{x}_t\mathbf{z}_{t-\tau}; \mathbf{z}_t) \\
&\stackrel{(P.2)}{=} \underbrace{I(\mathbf{z}_{t-\tau}; \mathbf{z}_t)}_{\text{Autoinformation}} + \underbrace{I(\mathbf{x}_t; \mathbf{z}_t|\mathbf{z}_{t-\tau})}_{\text{Superfluous Information}} . \quad (37)
\end{aligned}
$$

We can further factorize superfluous information by considering the immediate past $\mathbf{x}_{t-1}$ as follows:

$$
\begin{aligned}
\underbrace{I(\mathbf{x}_t; \mathbf{z}_t|\mathbf{z}_{t-\tau})}_{\text{Superfluous Information}} &\stackrel{(P.2)}{=} I(\mathbf{x}_t\mathbf{x}_{t-1}; \mathbf{z}_t|\mathbf{z}_{t-\tau}) - I(\mathbf{z}_{t-1}; \mathbf{z}_t|\mathbf{x}_t) \\
&\stackrel{(A.1)}{=} I(\mathbf{x}_t\mathbf{x}_{t-1}; \mathbf{z}_t|\mathbf{z}_{t-\tau}) \\
&\stackrel{(P.2)}{=} I(\mathbf{x}_{t-1}; \mathbf{z}_t|\mathbf{z}_{t-\tau}) + I(\mathbf{x}_t; \mathbf{z}_t|\mathbf{x}_{t-1}\mathbf{z}_{t-\tau}) \\
&\stackrel{(P.2)}{=} I(\mathbf{x}_{t-1}; \mathbf{z}_t|\mathbf{z}_{t-\tau}) + I(\mathbf{x}_t\mathbf{z}_{t-\tau}; \mathbf{z}_t|\mathbf{x}_{t-1}) - I(\mathbf{z}_{t-\tau}; \mathbf{z}_t|\mathbf{x}_{t-1}, \mathbf{x}_t) \\
&\stackrel{(A.1)}{=} I(\mathbf{x}_{t-1}; \mathbf{z}_t|\mathbf{z}_{t-\tau}) + I(\mathbf{x}_t\mathbf{z}_{t-\tau}; \mathbf{z}_t|\mathbf{x}_{t-1}) \\
&\stackrel{(P.2)}{=} I(\mathbf{x}_{t-1}; \mathbf{z}_t|\mathbf{z}_{t-\tau}) + I(\mathbf{x}_t; \mathbf{z}_t|\mathbf{x}_{t-1}) + I(\mathbf{z}_t; \mathbf{z}_{t-\tau}|\mathbf{x}_{t-1}) \\
&= \underbrace{I(\mathbf{x}_{t-1}; \mathbf{z}_t|\mathbf{z}_{t-\tau})}_{\text{Dynamic Information faster than } \tau} + \underbrace{I(\mathbf{x}_t; \mathbf{z}_t|\mathbf{x}_{t-1})}_{\text{Time-independent information}} , \quad (38)
\end{aligned}
$$

in which the last step follows from:

$$
0 \stackrel{(P.1)}{\leq} I(\mathbf{z}_t; \mathbf{z}_{t-\tau}|\mathbf{x}_{t-1}) \stackrel{(A.1)}{\leq} I(\mathbf{x}_t; \mathbf{x}_{t-\tau}|\mathbf{x}_{t-1}) \stackrel{(A.3)}{=} 0. \quad (39)
$$

Note that $I(\mathbf{x}_{t-1}; \mathbf{z}_t | \mathbf{z}_{t-\tau})$ refers to the information that $\mathbf{z}_t$ conveys about the immediate past $\mathbf{x}_{t-1}$ when the past representation $\mathbf{z}_{t-\tau}$ is observed. This quantity is positive whenever $\mathbf{z}_t$ contains information regarding processes that are faster than $\tau$, i.e. are not predictable from the past representation $\mathbf{z}_{t-\tau}$ but can be inferred from $\mathbf{z}_t$. The second term $I(\mathbf{x}_t; \mathbf{z}_t | \mathbf{x}_{t-1})$ refers to the information that $\mathbf{z}_t$ contains about processes that appear time-independent at the highest available time-resolution ($\tau = 1$). This component includes both time-independent noise and other time-dependent processes that appear uncorrelated at the observed temporal resolution. These two last components are indistinguishable without having access to higher-resolution sequences.

## C  COMPUTATION AND APPROXIMATIONS

### C.1  A TWO-STEP MINIMIZATION PROCEDURE

Consider the terms on the right side of expression 3. We use

$$\mathcal{L}^{LS}(\theta) := \mathrm{KL}(p([\mathbf{y}_{s+k\tau}]_{k=1}^{K} | \mathbf{x}_s) || p^{LS}([\mathbf{y}_{s+k\tau}]_{k=1}^{K} | \mathbf{x}_s)) \tag{40}$$

$$\mathcal{L}^{T}(\theta, \phi) := \sum_{k=1}^{K} \mathrm{KL}(p(\mathbf{z}_{s+k\tau} | \mathbf{z}_{s+(k-1)\tau}) || q_\phi(\mathbf{z}_{s+k\tau} | \mathbf{z}_{s+(k-1)\tau})) \tag{41}$$

$$\mathcal{L}^{P}(\theta, \psi) := \sum_{k=1}^{K} \mathrm{KL}(p(\mathbf{y}_{s+k\tau} | \mathbf{z}_{s+k\tau}) || q_\psi(\mathbf{y}_{s+k\tau} | \mathbf{z}_{s+k\tau})) \tag{42}$$

for notation brevity to underline the dependencies with the parameters $\theta$, $\phi$, $\psi$ for the encoder, variational transition, and variational predictive distributions respectively. The joint optimization can then be written as:

$$\min_{\theta, \phi, \psi} \mathcal{L}^{LS}(\theta) + \mathcal{L}^{T}(\theta, \phi) + \mathcal{L}^{P}(\theta, \psi) = \min_\theta \left[ \mathcal{L}^{LS}(\theta) + \min_\phi \mathcal{L}^{T}(\theta, \phi) + \min_\psi \mathcal{L}^{P}(\theta, \psi) \right]$$

$$\leq \mathcal{L}^{LS}(\hat{\theta}) + \min_\phi \mathcal{L}^{T}(\hat{\theta}, \phi) + \min_\psi \mathcal{L}^{P}(\hat{\theta}, \psi). \tag{43}$$

With $\hat{\theta} := \arg\min_\theta \mathcal{L}^{LS}(\theta)$.

The upper bound in equation 43 is still tight for flexible variational transition and prediction distribution. For a fixed $\hat{\theta}$, the variational transition and predictive gaps depend uniquely on the variational parameters $\phi$ and $\psi$ which can be optimized by minimizing the negative log-likelihood:

$$\arg\min_\phi \mathcal{L}^{T}(\hat{\theta}, \phi) = \arg\min_\phi \sum_{k=1}^{K} \mathbb{E}[-\log q_\phi(\mathbf{z}_{s+k\tau} | \mathbf{z}_{s+(k-1)\tau})] \tag{44}$$

$$\arg\min_\psi \mathcal{L}^{P}(\hat{\theta}, \psi) = \arg\min_\psi \sum_{k=1}^{K} \mathbb{E}[-\log q_\psi(\mathbf{y}_{s+k\tau} | \mathbf{z}_{s+k\tau})]. \tag{45}$$

### C.2  CONTRASTIVE LEARNING ON MARKOV PROCESSES

Consider the expression reported in equation 6:

$$\mathcal{L}_{\mathrm{InfoNCE}}^{\mathrm{T\text{-}InfoMax}}([\boldsymbol{x}_t]_{t=s}^{T}, \tau; \theta, \xi) := -\mathbb{E}\left[ \log \frac{e^{F_\xi(\boldsymbol{z}_t, \boldsymbol{z}_{t-\tau})}}{\mathbb{E}_{\boldsymbol{z}_t' \sim p(\mathbf{z}_t)}\left[ e^{F_\xi(\boldsymbol{z}_t', \boldsymbol{z}_{t-\tau})} \right]} \right] \tag{46}$$

$$\approx -\frac{1}{B} \sum_{i=1}^{B} \log \frac{e^{F_\xi(\boldsymbol{z}_{t_i}, \boldsymbol{z}_{t_i - \tau})}}{\frac{1}{B} \sum_{j=1}^{B} e^{F_\xi(\boldsymbol{z}_{t_j}, \boldsymbol{z}_{t_i - \tau})}}. \tag{47}$$

Focusing on the denominator in equation 46, we note that estimating the partition function would require sampling $\boldsymbol{z}_t'$ from $p(\mathbf{z}_t)$. If the dataset consists of multiple trajectories $\left[ \boldsymbol{x}_t^{(i)} \right]_{t=s_i}^{T_i} \overset{N}{\sim} p([\mathbf{x}_t]_{t=s}^{T})$, then this would require considering the representation of $\boldsymbol{x}_t^{(i)}$ from multiple trajectories at the given time $t$. Since we are considering time-independent homogeneous processes, even when the dataset

consists of a single trajectory $[\boldsymbol{x}_t]_{t=s}^T$, we can approximate samples from $p(\mathbf{x}_t)$ by considering any $\boldsymbol{x}_{t'}$ in the same sequence, with $t' \sim U(s, T)$. This approximation is accurate whenever $p(\mathbf{x}_t)$ approaches the equilibrium distribution and the trajectory $[\boldsymbol{x}_t]_{t=s}^T$ is long enough to obtain de-correlated samples. In case multiple trajectories are available at training time, this approach would benefit from creating mini-batches of inputs $\boldsymbol{x}_t^{(i)}$ (and corresponding representations $\boldsymbol{z}_t^{(i)}$) that are sampled from distinct trajectories:

$$\mathcal{L}_{\text{InfoNCE}}^{\text{T-InfoMax}}\left(\left\{\left[\boldsymbol{x}_t^{(i)}\right]_{t=s_i}^{T_i}\right\}_{i=1}^N, \tau; \theta, \xi\right) \approx -\frac{1}{B}\sum_{i=1}^B \log \frac{e^{F_\xi(\boldsymbol{z}_{t_i}^{(i)}, \boldsymbol{z}_{t_i-\tau}^{(i)})}}{\frac{1}{B}\sum_{j=1}^B e^{F_\xi(\boldsymbol{z}_{t_j}^{(j)}, \boldsymbol{z}_{t_i-\tau}^{(i)})}}. \tag{48}$$

## C.3 Superfluous information upper-bound

Computing superfluous information would require access to the true transition distribution $p(\mathbf{z}_t|\mathbf{z}_{t-\tau})$. Using standard variational inference, we can define a variational upper-bound based on the variational transition distribution instead:

$$\underbrace{I(\mathbf{x}_t; \mathbf{z}_t|\mathbf{z}_{t-\tau})}_{\text{Superfluous information}} = \mathbb{E}\left[\log \frac{p(\mathbf{z}_t|\mathbf{x}_t, \mathbf{z}_{t-\tau})}{p(\mathbf{z}_t|\mathbf{z}_{t-\tau})}\right]$$

$$= \mathbb{E}\left[\log \frac{p(\mathbf{z}_t|\mathbf{x}_t, \mathbf{z}_{t-\tau})}{q_\phi(\mathbf{z}_t|\mathbf{z}_{t-\tau})}\frac{q_\phi(\mathbf{z}_t|\mathbf{z}_{t-\tau})}{p(\mathbf{z}_t|\mathbf{z}_{t-\tau})}\right]$$

$$= KL(p_\theta(\mathbf{z}_t|\mathbf{x}_t)||q_\phi(\mathbf{z}_t|\mathbf{z}_{t-\tau})) - \underbrace{KL(p(\mathbf{z}_t|\mathbf{z}_{t-\tau})||q_\phi(\mathbf{z}_t|\mathbf{z}_{t-\tau}))}_{\text{Variational transition gap}}$$

$$\leq KL(p_\theta(\mathbf{z}_t|\mathbf{x}_t)||q_\phi(\mathbf{z}_t|\mathbf{z}_{t-\tau})). \tag{49}$$

The Expected value of KL-divergence between encoding and transition distribution can be estimated using sampled representations:

$$KL(p_\theta(\mathbf{z}_t|\mathbf{x}_t)||q_\phi(\mathbf{z}_t|\mathbf{z}_{t-\tau})) \approx \log \frac{p_\theta(\boldsymbol{z}_t|\boldsymbol{x}_t)}{q_\phi(\boldsymbol{z}_t|\boldsymbol{z}_{t-\tau})}, \tag{50}$$

with $\boldsymbol{z}_t$ and $\boldsymbol{z}_{t-\tau}$ as representations sampled from $p_\theta(\mathbf{z}_t|\boldsymbol{x}_t)$ and $p_\theta(\mathbf{z}_{t-\tau}|\boldsymbol{x}_{t-\tau})$ respectively, and $\boldsymbol{x}_t, \boldsymbol{x}_{t-\tau}$ as samples from the process at temporal distance $\tau$. Notably, this procedure is similar to the one used to enforce a bottleneck in Fischer (2020).

# D Comparison with the literature

## D.1 Linear Correlation Maximization and Mutual Information

A conventional and successful approach to mutual information maximization is the maximization of linear autocorrelation (Andrew et al., 2013; Noé & Nüske, 2013). This can be expressed as:

$$\arg\max_\theta Tr(\text{Cov}[\mathbf{z}_{t-\tau}, \mathbf{z}_t]) \text{ subject to } \text{Cov}[\mathbf{z}_{t-\tau}, \mathbf{z}_{t-\tau}] = \text{Cov}[\mathbf{z}_t, \mathbf{z}_t] = \mathbf{I} \tag{51}$$

Here the maximization of the covariance trace is equivalent to the maximization of the sum of its $D$ eigenvalues $\lambda_i$, where $D$ denotes the dimensionality of the representation.

A variety of surrogates maximize the sum of squared eigenvalues (Mardt et al., 2018; Wu & Noé, 2020) or the squared Euclidean distance in the representation space (Lyu et al., 2022; Wiskott & Sejnowski, 2002). This objective can also be equivalently interpreted as maximizing mutual information for jointly Normal random variables (Borga, 2001) with linear encoders.

Assume that the representations $\mathbf{z}_{t-\tau}$ and $\mathbf{z}_t$ are jointly Normal distributed:

$$[\mathbf{z}_{t-\tau}, \mathbf{z}_t] \sim \mathcal{N}(\boldsymbol{\mu}, \boldsymbol{S}) \text{ with } \boldsymbol{S} = \begin{bmatrix} \boldsymbol{S}_{t-\tau,t-\tau} & \boldsymbol{S}_{t-\tau,t} \\ \boldsymbol{S}_{t,t-\tau} & \boldsymbol{S}_{t,t} \end{bmatrix} \tag{52}$$

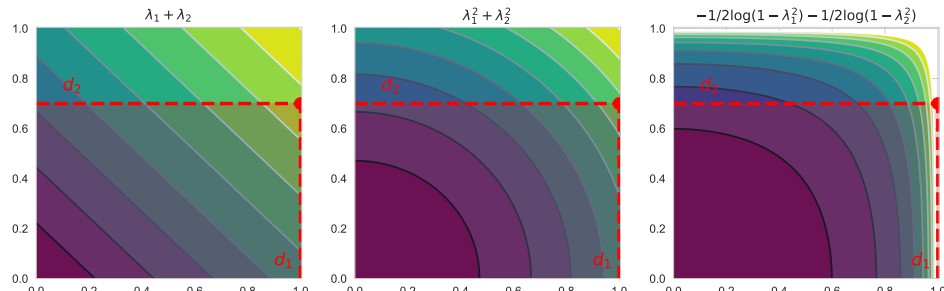

Figure 6: Visualization of several objectives as a function of the eigenvalues $\lambda_1$ and $\lambda_2$ of $\boldsymbol{S}_{t,t-\tau}\boldsymbol{S}_{t-\tau,t}$. The vertical lines for $d_1$ and $d_2$ correspond to the eigenvalues of $\boldsymbol{\Sigma}_{t,t-\tau}\boldsymbol{\Sigma}_{t-\tau,t}$ determined by the original covariance $\boldsymbol{\Sigma}_{t-\tau,t}$. Note that whenever $\mathbf{z}_t$ is a linear projection of $\mathbf{x}_t$, $\lambda_1$ and $\lambda_2$ are constrained to be in the shaded region determined by $d_1$ and $d_2$. As a result, all objectives are optimal for $\lambda_1 = d_1$ and $\lambda_2 = d_2$, which corresponds to a projection onto the principal components.

In this instance, autoinformation can be directly computed as follows:

$$
\begin{aligned}
AI_{\mathcal{N}}(\mathbf{z}_{t-\tau}, \tau) &= \frac{1}{2} \log \frac{\det \boldsymbol{S}_{t-\tau,t-\tau} \det \boldsymbol{S}_{t,t}}{\det \boldsymbol{S}} \\
&= \frac{1}{2} \log \frac{\det \boldsymbol{S}_{t,t}}{\det \left( \boldsymbol{S}_{t,t} - \boldsymbol{S}_{t,t-\tau} \boldsymbol{S}_{t-\tau,t-\tau}^{-1} \boldsymbol{S}_{t-\tau,t} \right)} \\
&= -\frac{1}{2} \log \det \left( \mathbf{I} - \boldsymbol{A} \right) \\
&= -\frac{1}{2} \log \det \left( \boldsymbol{U}(\mathbf{I} - \boldsymbol{\Lambda})\boldsymbol{U}^T \right) \\
&= -\frac{1}{2} \log \det \left( \mathbf{I} - \boldsymbol{\Lambda} \right)) \\
&= -\frac{1}{2} \sum_{i=1}^{D} \log \left( 1 - \lambda_i \right)
\end{aligned}
\tag{53}
$$

In which $\boldsymbol{A} := \boldsymbol{S}_{t,t}^{-1/2} \boldsymbol{S}_{t,t-\tau} \boldsymbol{S}_{t-\tau,t-\tau}^{-1} \boldsymbol{S}_{t-\tau,t} \boldsymbol{S}_{t,t}^{-1/2}$, and $\boldsymbol{U}\boldsymbol{\Lambda}\boldsymbol{U}^T$ refers to its eigendecomposition, and $\lambda_i$ the corresponding eigenvalues. Under the assumption that $\boldsymbol{S}_{t-\tau,t-\tau}$ and $\boldsymbol{S}_{t,t}$ are restricted to be identity matrices, the expression for $\boldsymbol{A}$ simplifies to $\boldsymbol{A} = \boldsymbol{S}_{t,t-\tau} \boldsymbol{S}_{t-\tau,t}$.

As illustrated in Figure 6, for any linear encoder in the form $\mathbf{z}_t = \boldsymbol{W}\mathbf{x}_t$, maximizing the sum of the eigenvalues of $\boldsymbol{A}$, the sum of their squared values, or the expression in equation 53 is equivalent. This is true because under the constraint $\boldsymbol{S}_{t,t} = \boldsymbol{S}_{t-\tau,t-\tau} = \mathbf{I}$, the eigenvalues of $\boldsymbol{S}_{t,t-\tau} \boldsymbol{S}_{t-\tau,t}$ are upper-bounded by the eigenvalues of $\boldsymbol{\Sigma}_{t,t-\tau} \boldsymbol{\Sigma}_{t-\tau,t}$, with $\boldsymbol{\Sigma}_{t-\tau,t} := \text{Cov}[\mathbf{x}_{t-\tau}, \mathbf{x}_t]$.

Note that although the correlation matrix does capture linear relation between $\mathbf{z}_{t-\tau}$ and $\mathbf{z}_t$, it does not consider higher-order interaction between the representations. This is a limiting factor especially for low-dimensional representations because of the expressive power of linear transformations. This phenomenon can be clearly observed by comparing the autoinformation plots in Figure 4b (2D representations) and Figure 13 (16/32 dimensional representations). The autoinformation extracted by representations that use linear correlation maximization (TICA and VAMPNet) strongly depends on the number of dimensions of the representation $\mathbf{z}_t$. The effect on methods that rely on non-linear contrastive mutual information maximization methods is much more moderate, making them more flexible and suitable for 2D visualizations.

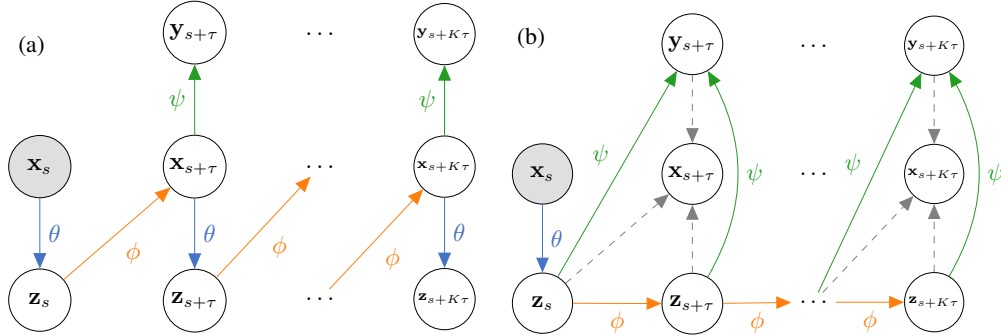

Figure 7: Viable inference schemes for maximal future state information: $AI(\mathbf{x}_t; \tau) - I(\mathbf{z}_t; \mathbf{x}_{t+\tau}) = 0$. The difference with the Latent Simulation inference scheme lies in the lack of the conditional independence $I(\mathbf{z}_t; \mathbf{x}_{t+\tau}|\mathbf{z}_{t+\tau}) = 0$. Note that modeling $q_\phi(\mathbf{x}_t|\mathbf{z}_{t-\tau})$ is generally more difficult than modeling latent transitions $q_\phi(\mathbf{z}_t|\mathbf{z}_{t-\tau})$.

### D.2 Maximizing information with respect to future states

Several models in the literature consider the reconstruction of future states as the training objective (Wehmeyer & Noé, 2018; Wang et al., 2019b). This objective can be interpreted as the maximization of the mutual information between the current representation $\mathbf{z}_t$ and the future state $\mathbf{x}_{t+\tau}$ Poole et al. (2019):

$$\max_\theta I(\mathbf{z}_t; \mathbf{x}_{t+\tau}) = \max_\theta H(\mathbf{x}_{t+\tau}) - H(\mathbf{x}_{t+\tau}|\mathbf{z}_t)$$
$$\geq H(\mathbf{x}_{t+\tau}) - \min_{\theta,\phi} \mathbb{E}_{p(\mathbf{x}_t)p_\theta(\mathbf{z}_t|\mathbf{z}_t)}\left[-\log q_\phi(\mathbf{x}_{t+\tau}|\mathbf{z}_t)\right], \tag{54}$$

in which $q_\phi(\mathbf{x}_{t+\tau}|\mathbf{z}_t)$ refers to the decoder that predicts the future states given the current representation.

We note that autoinformation in $\mathbf{z}_t$ is always smaller or equal to $I(\mathbf{z}_t; \mathbf{x}_{t+\tau})$, which we will refer to as *future predictive information*:

$$AI(\mathbf{x}_t; \tau) = I(\mathbf{x}_t; \mathbf{x}_{t+\tau}) \geq I(\mathbf{z}_t; \mathbf{x}_{t+\tau}) \geq I(\mathbf{z}_t; \mathbf{z}_{t+\tau}) = AI(\mathbf{z}_t; \tau). \tag{55}$$

Preserving autoinformation is a stronger condition than having maximal future predictive information:

$$AIG(\mathbf{z}_t; \tau) = 0 \implies AI(\mathbf{x}_t; \tau) - I(\mathbf{z}_t; \mathbf{x}_{t+\tau}) = 0. \tag{56}$$

This is because the additional condition $I(\mathbf{z}_t; \mathbf{x}_{t+\tau}|\mathbf{z}_{t+\tau}) = 0$ is required:

$$AIG(\mathbf{z}_t; \tau) = AI(\mathbf{x}_t; \tau) - I(\mathbf{z}_t; \mathbf{z}_{t+\tau})$$
$$= \underbrace{AI(\mathbf{x}_t; \tau) - I(\mathbf{z}_t; \mathbf{x}_{t+\tau})}_{\text{Missing future predictive information}} + I(\mathbf{z}_t; \mathbf{x}_{t+\tau}|\mathbf{z}_{t+\tau}). \tag{57}$$

This additional condition is not directly required to prove the results of Lemma 1, Lemma 2, and Lemma 3, which can be extended to the condition $AI(\mathbf{x}_t; \tau) - I(\mathbf{z}_t; \mathbf{x}_{t+\tau}) = 0$. However, the lack of the conditional independence $I(\mathbf{z}_t; \mathbf{x}_{t+\tau}|\mathbf{z}_{t+\tau}) = 0$ would result in a difference inference scheme, in which instead of approximating transitions directly in the latent space, each step would require modeling transitions from $\mathbf{z}_t$ to $\mathbf{x}_{t+\tau}$, as shown in Figure 7a. Alternatively, one could model latent transitions $q_\phi(\mathbf{z}_t|\mathbf{z}_{t-\tau})$, but the predictive target distribution would depend on both the current and future representation, as shown in Figure 7b. Both inference schemes and the maximization of $I(\mathbf{z}_t; \mathbf{x}_{t+\tau})$ are more computationally expensive than the proposed T-InfoMax training procedure and Latent Simulation inference.

### D.3 State Predictive Information Bottleneck and target sufficiency

Wang & Tiwary (2021) introduce a State Predictive Information Bottleneck (SPIB) objective aiming to create a representation $\mathbf{z}_t$ that is sufficient for the next target $\mathbf{y}_{t+\tau}$ while compressing information:

$$\mathcal{L}^{SPIB}(\theta; \beta, \tau) = -\mathbb{E}_t[I(\mathbf{z}_t; \mathbf{y}_{t+\tau}) - \beta I(\mathbf{x}_t; \mathbf{z}_t)]. \tag{58}$$

Although this objective seems natural for training effective representations, we can show that sufficiency for a given target $\mathbf{y}_{t+\tau}$ is a necessary but not sufficient condition for autoinformation preservation. As a result, a representation that is optimal according to the SPIB objective may introduce inference error even when the true latent transition $p(\mathbf{z}_t|\mathbf{z}_{t-\tau})$ and latent future predictive $p(\mathbf{y}_{t+\tau}|\mathbf{z}_t)$ distributions are available at inference time, as shown in the following example.

Consider a dynamic system in which each state is described by a particle position, velocity, and acceleration governed by a simple time-discrete update:

$$\mathbf{x}_t = \begin{bmatrix} \mathbf{r}_t \\ \mathbf{v}_t \\ \mathbf{a}_t \end{bmatrix} = \begin{bmatrix} \mathbf{r}_{t-\tau} + \tau\mathbf{v}_{t-\tau} \\ \mathbf{v}_{t-\tau} + \tau\mathbf{a}_{t-\tau} \\ \boldsymbol{\eta}_t \end{bmatrix} = D_\tau(\mathbf{x}_{t-\tau}, \boldsymbol{\eta}_t), \tag{59}$$

in which the acceleration at each time step is sampled from a time-independent Normal distribution $\boldsymbol{\eta}_t \sim \mathcal{N}(\mathbf{0}, \mathbf{1})$ and $D_\tau$ refers to the function used to unroll the true system dynamics at the time scale $\tau$. Clearly, the system is an instance of a homogenous Markov process.

We are interested in predicting the particle position $\mathbf{y}_t = \mathbf{r}_t$. Clearly, since the next position depends solely on the current position and the current velocity, we have that a representation that contains only velocity and position information is sufficient for the next target prediction:

$$I(\mathbf{y}_{t+\tau}; \mathbf{x}_t) = I(\mathbf{y}_{t+\tau}; \mathbf{z}_t^{SPIB}) \quad \text{with } \mathbf{z}_t^{SPIB} = \begin{bmatrix} \mathbf{r}_t \\ \mathbf{v}_t \end{bmatrix} \tag{60}$$

On the other hand, a representation that maximizes autoinformation (and is optimal according to Equation 8) must also contain information regarding the acceleration since current acceleration is predictive for the future velocity:

$$I(\mathbf{x}_t; \mathbf{x}_{t+\tau}) = I(\mathbf{z}_t^{T-IB}; \mathbf{z}_{t+\tau}^{T-IB}) > I(\mathbf{z}_t^{SPIB}; \mathbf{z}_{t+\tau}^{SPIB}) \quad \text{with } \mathbf{z}_t^{T-IB} = \begin{bmatrix} \mathbf{r}_t \\ \mathbf{v}_t \\ \mathbf{a}_t \end{bmatrix}. \tag{61}$$

Note that a representation that is optimal according to SPIB would instead explicitly discard acceleration because of the compression regularization:

$$I(\mathbf{x}_t; \mathbf{z}_t^{T-IB}) > I(\mathbf{x}_t; \mathbf{z}_t^{SPIB}). \tag{62}$$

Since both representations are sufficient for $\mathbf{y}_{t+\tau}$, they yield the same predictive distribution for the next target:

$$p(\mathbf{y}_{t+\tau}|\mathbf{z}_t^{SPIB}) = p(\mathbf{y}_{t+\tau}|\mathbf{z}_t^{T-IB}) = p(\mathbf{y}_{t+\tau}|\mathbf{x}_t). \tag{63}$$

However, if we look at the predictive distribution at times larger than $\tau$, we observe some discrepancies. In particular, we can show that:

$$p(\mathbf{y}_{t+2\tau}|\mathbf{z}_t^{SPIB}) \neq p(\mathbf{y}_{t+2\tau}|\mathbf{z}_t^{T-IB}) = p(\mathbf{y}_{t+2\tau}|\mathbf{x}_t), \tag{64}$$

In which the first inequality follows from the fact that $\mathbf{z}_t^{SPIB}$ does not contain knowledge about the acceleration, while the second inequality follows from Lemma 1+Lemma 3. Therefore we showed that latent simulation performed on representations that are optimal according to the SPIB objective (and not according to T-IB) introduces inference error for time scales larger than $\tau$. The intuition is that sufficiency for the next target $\mathbf{y}_{t+\tau}$ does not guarantee a transfer of the Markov property from the original space $\mathbf{x}_t$ to the representation $\mathbf{z}_t$. That requirement is satisfied only whenever the representation $\mathbf{z}_t$ preserves autoinformation, as shown in Lemma 1+ Lemma 2.

# E EXPERIMENTAL DETAILS

We include additional details regarding the training data, architectures, and optimization procedure to ensure the reproducibility of the reported results.

## E.1 DATA

### E.1.1 PRINZ 2D

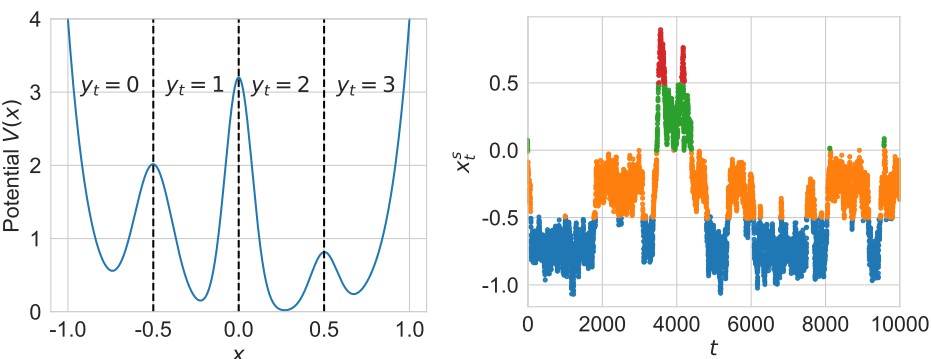

Figure 8: Left: visualization of the 1D Prinz potential, and the corresponding regions used to define the discrete targets $\boldsymbol{y}$. Right: Visualization of the 1D slow component $x_t^s$ colored by $y_t^s$.

The Prinz 2D trajectories consist of sequences of 100K data points generated by diffusing a point particle into a potential $V(x) := 4\left(x^8 + 0.8e^{-80x^2} + 0.2e^{-80(x-0.5)^2} + 0.5e^{-40(x+0.5)^2}\right)$ with an Euler-Maruyama integrator following the update:

$$x_{t+1} = x_t - h\nabla V(x_t) + \sqrt{h}\,\eta_t, \tag{65}$$

in which $h = 10^{-4}$ refers to the integrator step and $\eta_t$ is standard Normal uncorrelated noise. We generate $\left[x_t^f\right]_{t=s}^T$ by performing 160 integration steps in-between consecutive timesteps, while $[x_t^s]_{t=s}^T$ is generated by considering 5 integration steps. The Deep Time package (Hoffmann et al., 2021) is used to produce the slow and fast trajectories, and the corresponding potential $V(x)$ is visualized in Figure 8. The fast and slow independent components are then mixed as follows:

$$\boldsymbol{x}_t = \begin{bmatrix} \tanh(x_t^s + x_t^f) \\ \tanh(x_t^s - x_t^f) \end{bmatrix}, \tag{66}$$

to produce the trajectories visualized in Figure 3a.

### E.1.2 MOLECULAR DATA

**Trajectories** We analyze trajectories obtained by simulating *Alanine Dipeptide*, *Chignolin*, and *Villin* (Lindorff-Larsen et al., 2011). For Alanine Dipeptide, the three splits correspond to separate simulations of 250K/100K/100K frames respectively. In contrast, for Chignolin and Villin simulation, a single trajectory is split into 3 temporally disjoint parts: 334.743/100K/100K frames for Chignolin, and 427.907/100K/100K frames for Villin. Each observation $\mathbf{x}_t$ consists of the set of the Euclidean coordinates of all the atoms and a one-hot corresponding to the atomic number for the Alanine Dipeptide trajectories. The input data for the mini-proteins, on the other hand, consists of a coarse-grained representation indicating the 3D location of the amino acids in the protein chain (10 for Chignolin and 35 for Villin), along with a one-hot encoding for the amino acid type.

**Targets** Targets for the Alanine Dipeptide molecules are generated by clustering torsion angles $\phi$ and $\psi$ into 6 regions, corresponding to the known meta-stable states. For the Chignolin and Villin molecules, we generate targets $\mathbf{y}_t$ by clustering the 32D invariant TICA projections obtained by

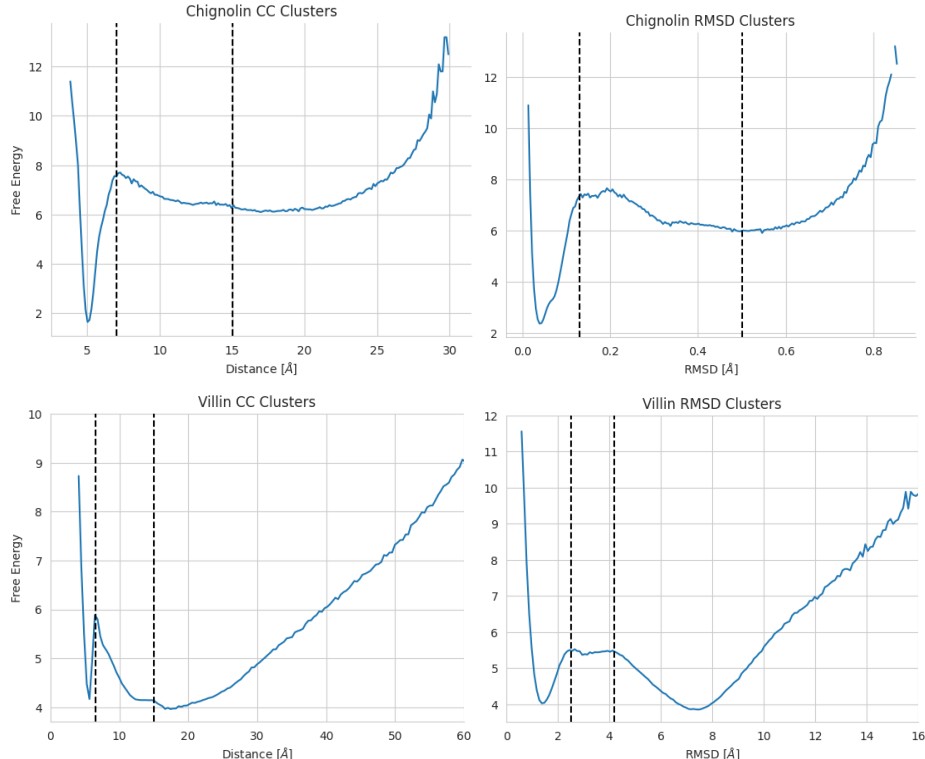

Figure 9: Visualization of the 1D free energy induced by the distribution of the distances between first and last C-alpha atoms in the chain (CC) and Root Mean Squared Distance (RMSD) for the molecules of Chignoling and Villin. Vertical dashed lines are used to denote the margin between the different discretized regions. The thresholds are set to $[7, 15]$, $[1.3, 5]$ Angstroms for Chignolin CC and RMDS respectively, while the values of $[6.5, 15]$ and $[2.5, 4.2]$ are used for Villin.

following the same procedure described in Köhler et al. (2023) using KMeans with 5 centroids, as depicted in Figure 4a. We produce additional sets of targets by considering the distance between the first and last C-alpha carbon atoms in the amino acid sequence (CC, 3 clusters), and the Root Mean Squared Distance (RMSD, 3 clusters) from the stable folded configuration. The thresholds used to create the clusters are visualized together with the corresponding free energy in Figure 9.

**Lag time** We decide on a training lag time $\tau$ for each molecule that is long enough to capture relevant meta-stable state transitions, see Figure 4b and Figure 13. We focus on a time scale on which most of the dynamic information is still present while modeling transitions that are orders of magnitudes larger compared to the original simulations. We used a train lag time of 16 ps on Alanine Dipeptide simulations, while 3200 ps was used for the Chignolin and Villin simulations. The same value of $\tau$ is used both to train the encoder (step 1) and the transition model (step 2).

## E.2 ARCHITECTURES AND OPTIMIZATION

The models used for the experiments reported in this paper are described in detail in the following sections. The experiments reported in this paper required a total of 25 days of computation on $A100$ GPUs. This estimation includes model development, hyper-parameter tuning, and evaluation.

### E.2.1 ENCODER TRAINING

We train each encoder for a maximum of 50 epochs with mini-batches of size 512 using the AdamW (Loshchilov & Hutter, 2019) optimizer. To prevent overfitting, we use early stopping based on the validation loss. Following previous work (Chen et al., 2020), the models are trained with an initial learning rate of $10^{-6}$, which is gradually increased up to $5 \times 10^{-4}$ over the course of 5 epochs with a

linear schedule. The learning rate is then decreased to the initial value using a cosine schedule over the following 45 epochs.

For the Prinz 2D experiments, encoders consist of MLPs with two hidden units of size 64 and a 2D output. In the molecular settings, each encoder architecture consists of a TorchMD Equivariant Transformer (Thölke & Fabritiis, 2022) followed by global mean pooling and a linear layer to produce a rotation, translation, reflection, and permutation invariant representation for each molecule. We use a total of 3 layers of 32 hidden units with 8 heads each for the Alanine Dipeptide experiment. The more challenging Chignolin and Villin molecules use encoders consisting of 5 layers with 64 hidden units and 8 projection heads instead. For the evaluation of the quality of unfolded trajectories, we use a 16-dimensional representation for Alanine Dipeptide. A total of 32 dimensions are used for Chignolin and Villin.

**TICA** Temporal Independent Component Analysis consists of a linear projection of the input data onto the principal temporal components. As a result, $p_\theta(\mathbf{z}_t|\mathbf{x}_t)$ consists of a simple linear projection instead of a neural network that has been optimized using the Deeptime python library (Hoffmann et al., 2021). For the Prinz2D experiments, we apply TICA directly to the original sequence $[\mathbf{x}_t]_{t=s}^{T}$ to project each data point $\mathbf{x}_t$ onto the principal temporal component $\mathbf{z}_t$. For the Alanine Dipeptide Experiments, the TICA representations correspond directly to the torsion angles determined by the carbon skeleton (2 angles), which are commonly used in literature to describe the configuration of this small molecule (Vymětal & Vondrášek, 2010; Mardt et al., 2018). The representations for Chignolin and Villin are produced following the same procedure described in detail in Köhler et al. (2023), in which torsion angles and inter-atomic distances are projected onto the principal temporal components.

**VAMPNet** We train the encoder $p_\theta(\mathbf{z}_t|\mathbf{x}_t)$ using VAMP-2 score (Mardt et al., 2018; Wu & Noé, 2020) using the implementation from the Deeptime python library (Hoffmann et al., 2021). The VAMPNet model was originally designed for estimating dominant spectral components of molecular simulations. However, Sidky et al. (2020) has shown the effectiveness of VAMPNet for Latent Simulation inference.

**T-InfoMax** As a representative of non-linear mutual information maximization methods, we consider the popular InfoNCE method (van den Oord et al., 2018; Chen et al., 2020). Following the literature (van den Oord et al., 2018; Poole et al., 2019), we model the log-ratio between joint and product distribution with a separable architecture:

$$F_\xi(\mathbf{z}_{t-\tau}; \mathbf{z}_t) = g_{\xi_1}(\mathbf{z}_{t-\tau})^T g_{\xi_2}(\mathbf{z}_t), \tag{67}$$

in which $g_{\xi_1}$ and $g_{\xi_2}$ are neural networks mapping the latent representations into a 128-dimensional normalized vector. The two architectures have distinct weights with one hidden layer of 256 units and group normalization (Wu & He, 2018) before the ReLU non-linearity.

**T-IB** Analogously to the T-InfoMax counterpart, the Time-lagged Information Bottleneck objective makes use of InfoNCE for time-lagged information maximization, with an additional regularization term modulated by the hyper-parameter $\beta$ as shown in equation 9. Following (Federici et al., 2020) we first train the encoder with an initial value of $\beta = 10^{-6}$ for 5 epochs. This is to prevent the representation from collapsing into a constant at the beginning of training. Secondly, the regularization strength is gradually increased up to the final desired value over the course of 30 epochs. We empirically observed that the T-IB models benefit from the use of a stochastic encoder $p_\theta(\mathbf{z}_t|\mathbf{x}_t) = \mathcal{N}(\mathbf{z}_t|\mu_\theta(\mathbf{x}_t), \sigma_\theta(\mathbf{x}_t)\mathbf{I})$. The parameter vectors $\mu_\theta(\mathbf{x}_t)$ and $\sigma_\theta(\mathbf{x}_t)$ are obtained using two linear projection heads on top of the encoder features, as a result, the size of the stochastic encoders is comparable to the corresponding deterministic counterpart.

We initialize the architectures with a value of $\sigma_\theta(\mathbf{x}_t) \approx 10^{-4}$ to reduce the amount of Gaussian additive noise in the initial part of the training. Empirical results showed that the additional stochasticity produces smooth transitions between different levels of regularization strength. This is in contrast with the sharp regime changes observed with deterministic encoders (Figure 11, **Top**). We believe that this is due to the fact that the addition of Gaussian noise allows the encoder to destroy superfluous information locally when necessary.

### E.2.2 Transition and Prediction training

The variational transition and prediction models ($q_\phi(\mathbf{z}_t|\mathbf{z}_{t-\tau})$ and $q_\psi(\mathbf{z}_t|\mathbf{y}_t)$, respectively) are jointly trained on the embedding produced by the encoder trained in the previous step. The training procedure uses mini-batches of size 512 with AdamW and a fixed learning rate of $5 \times 10^{-4}$ over a total of 50 epochs. Contrary to the previous step, we did not observe any overfitting with only marginal improvements in the training and validation scores by the end of the training procedure.

**Transition** The transition model consists of conditional Flow++ layers (Ho et al., 2019) due to their flexibility, sampling speed, and ability to model correlated distributions. The transitions for Prinz2D and Alanine Dipeptide representations consist of 3 flow layers. Each layer is composed of a conditional mixture of logistics CDF coupling transformation consisting of a neural network with two hidden layers of 64 hidden units, which maps the representations $\mathbf{z}_{t-\tau}$ into the parameters of a mixture of 16 logistics distributions. An architecture of 5 layers is used to learn the more challenging transition distributions for Chignolin and Villin. To prevent numerical overflows while unfolding long simulations, we clip samples to be in the interval $[-10^6, 10^6]$.

**Prediction** Each feature predictor used in this work consists of a simple 1-hidden layer MLP with 128 hidden units mapping the representation $\mathbf{z}_t$ into the logits for the variational predictive distribution $q_\psi(\mathbf{y}_t|\mathbf{z}_t)$.

### E.3 Evaluation

We focus our evaluation on two main aspects. First, we analyze the amount of autoinformation that several models extract from the molecular data to better understand which temporal characteristics of the molecular process are successfully captured. The second aspect involves the evaluation of the fidelity of trajectories unfolded using the Variational Latent Simulation process.

#### E.3.1 Mutual Information

**Autoinformation** We estimate autoinformation for evaluation purposes using SMILE (Song & Ermon, 2020) on the trained representations $\mathbf{z}_t$ with a clipping interval of $[-5, 5]$. The ratio estimation architecture consists of an initial projection head $g : \mathbb{Z} \to \mathbb{R}^{128}$ with one hidden layer of 256 units and output $\mathbf{h}_t := g(\mathbf{z}_t)$ with a dimension of 128. Pairs of the 128-dimensional feature vectors $\mathbf{h}_t$ at different temporal resolutions are then concatenated and fed into a second MLP $r_\tau : \mathbb{R}^{128} \times \mathbb{R}^{128} \to \mathbb{R}$ with 64 hidden units and 1 output, which corresponds to the estimated log-ratio value. Each pair of $\mathbf{h}_t, \mathbf{h}_{t+\tau}$ is fed into a distinct architecture $r_\tau$ for each $\tau$. This setup allows us to estimate autoinformation at several time-lags at once to produce the plots visualized in Figure 4b, Figure 13 and Figure 14a. Each dot in the figure corresponds to the expected output of one ratio estimation model $r_\tau(g(\mathbf{z}_t), g(\mathbf{z}_{t+\tau}))$ on the entirety of the training set. The ratio estimation models are fit for at most 20 epochs using early stopping based on the validation loss. Note that samples from the marginal distribution used to estimate the value of the partition function are sampled by sampling $\mathbf{x}_{t'}$ with uniform probability using the same strategy described in Appendix C.2. Estimation is performed using the Torch-Mist package(Federici et al., 2023).

**Target Information** Following Poole et al. (2019); McAllester & Stratos (2020); Song & Ermon (2020), we estimate the amount of target information in the representations as a difference of cross-entropies:

$$I(\mathbf{z}_t; \mathbf{y}_t) = H(\mathbf{y}_t) - H(\mathbf{y}_t|\mathbf{z}_t) \leq H(\mathbf{y}_t) - \mathbb{E}[-\log q_\psi(\mathbf{y}_t|\mathbf{z}_t)], \tag{68}$$

in which the marginal entropy $H(\mathbf{y}_t)$ for the discrete targets $\mathbf{y}_t$ is estimated by counting the frequency of each class, while the expected cross entropy $\mathbb{E}[-\log q_\psi(\mathbf{y}_t|\mathbf{z}_t)]$ is evaluated using the trained predictor $q_\psi(\mathbf{y}_t|\mathbf{z}_t)$ on the entirety of the test trajectory and computing the corresponding expected log-likelihood. Note that with $I(\mathbf{z}_t; \mathbf{y}_t)$ we implicitly refer to the expected amount of target information over an entire trajectory rather than the amount of information estimated specifically at the time-step $t$.

#### E.3.2 Unfolding trajectories

Accurately estimating a measure of divergence between joint distributions when only samples are accessible is generally a challenging task due to the number of samples required for a reliable

estimation. For this reason, instead of considering continuous multi-dimensional targets $\mathbf{y}_t$, we focus our attention on discrete targets. The targets in our experiments are designed to capture properties of interest of the trajectories

Our evaluation procedure can be described in 3 steps:

1. First we encode the initial (unobserved) test state $\boldsymbol{x}_s$ into the latent configuration $\boldsymbol{z}_s$ using $p_\theta(\mathbf{z}_t|\mathbf{x}_t)$. Starting from $\boldsymbol{z}_s$, we sample a total of 256 trajectories $\left[\tilde{\boldsymbol{z}}_{s+k\tau}^{(i)}\right]_{k=1}^K$ by sampling from the variational transition model $q_\phi(\mathbf{z}_t|\mathbf{z}_{t-\tau})$ sequentially for a total temporal duration which is comparable to the time-span covered by the test trajectories $T - s \approx K\tau$. Using the prediction model $q_\psi(\mathbf{y}_t|\mathbf{z}_t)$ we then sample a target $\tilde{y}_t^{(i)}$ for each sampled $\tilde{\boldsymbol{z}}_t^{(i)}$, obtaining 256 sequences of targets $\left[\tilde{y}_{+k\tau}^{(i)}\right]_{k=1}^K$.

2. We count the number of transitions from each discrete state $\tilde{y}_t^{(i)}$ to the following $\tilde{y}_{t+k\tau}^{(i)}$ for various numbers of steps $k$, effectively creating a series of transition count matrix $\tilde{\boldsymbol{C}}_{k\tau}^{(i)}$ and $\boldsymbol{C}_{k\tau}$ respectively for $\left[\tilde{y}_{+k\tau}^{(i)}\right]_{k=1}^K$ and $[y_t]_{t=s}^T$. The 256 count matrices for the unfolded trajectories are then averaged to produce $\tilde{\boldsymbol{C}}_{k\tau} = 1/256 \sum_{i=1}^{256} \tilde{\boldsymbol{C}}_{k\tau}^{(i)}$. We normalize each row of $\tilde{\boldsymbol{C}}_{k\tau}$ and $\boldsymbol{C}_{k\tau}$ to estimate the transition probability matrices $\tilde{\boldsymbol{T}}_{k\tau}$ and $\boldsymbol{T}_{k\tau}$. Analogously, we count the number of times that each state is visited to determine the normalized counts $\boldsymbol{m}$ and $\tilde{\boldsymbol{m}}$ using the ground truth and unfolded trajectories respectively.

3. We compute the Jensen-Shannon divergence between each row of $\boldsymbol{T}_{k\tau}$ and $\tilde{\boldsymbol{T}}_{k\tau}$, then we average the values obtained for each row into a single number, representing the average Jensen-Shannon divergence. With this last step, we obtain one value of transition Jensen-Shannon divergence ($TJS$) for each chosen number of unfolding steps $k$ (see Figure 15). The values for each row are averaged using the same weighting instead of the relative state probability to accentuate errors when transitioning from rare states. Analogously we compute the value of marginal JS ($MJS$) by computing the divergence between the probability distribution induced by $\boldsymbol{m}$ and $\tilde{\boldsymbol{m}}$.

## F  ADDITIONAL RESULTS

In this section, we report additional ablation studies and the performance of the models considered in this analysis for different sets of targets.

### F.1  T-IB REGULARIZATION STRENGTH AND TRAIN LAG TIME

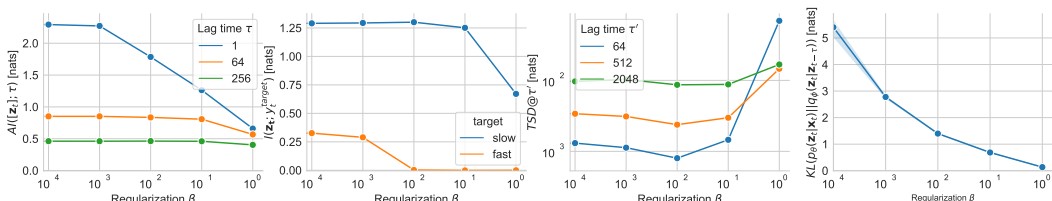

Figure 10: Visualization of the effect of the regularization strength $\beta$ on Autoinformation, information regarding slow and fast modes, transition $JS$, and amount of superfluous information on the Prinz 2D dataset. All representations are trained using $\tau = 64$. Representations trained with $\beta < 0.01$ tend to contain information regarding the fast mode and higher autoinformation at small temporal scales, while strong regularization $\beta > 0.1$ results in representations that contain too little information. Note that the best performance in terms of transition $JS$ divergence is achieved by the representation that contains the least information regarding $\mathbf{y}_t^f$ and most about $\mathbf{y}_t^s$, which corresponds to the most compressed sufficient representation.

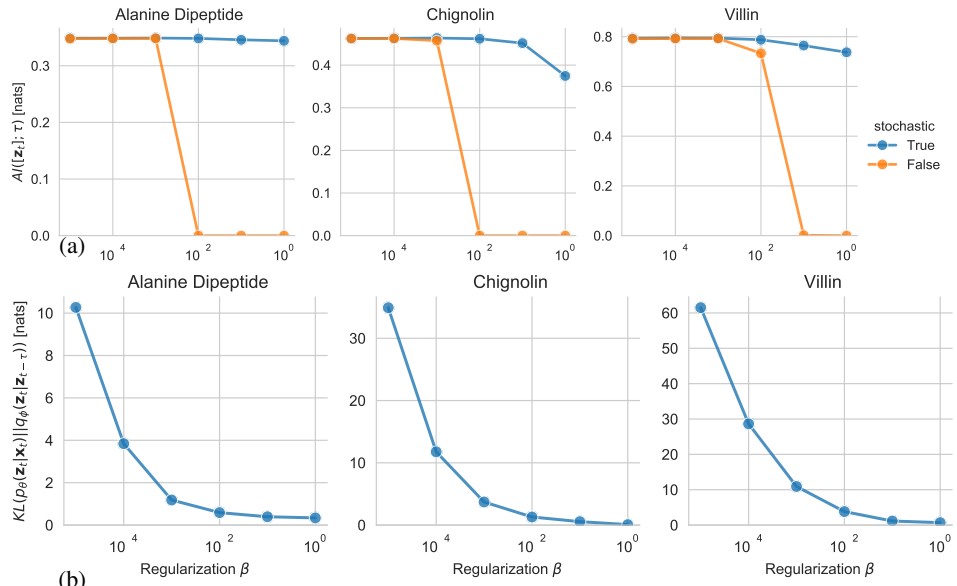

Figure 11: Visualization of the effect of the regularization strength on the representations produced with T-IB on molecular simulations with fixed train lag time $\tau$. 11a: estimated autoinformation (y-axis) for the three molecules as a function of the training regularization strength $\beta$ (x-axis). Stochastic encoders (in blue) show a much smoother interpolation. 11b: the amount of superfluous information (y-axis, Equation 49) as a function of the regularization strength.

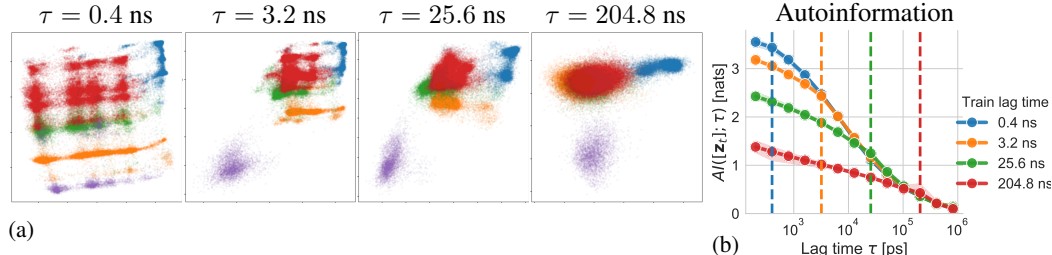

Figure 12: Visualization of the effect of the train lag time $\tau$ on 2D T-IB representations of Villin trained with $\beta = 0.01$. 12a: representation of the test trajectories for models trained with several lag times, colored by the clustered TICA embedding, as reported in Figure 4a. 12b: corresponding autoinformation plot in which the dashed vertical lines correspond to the respective training lag times. Note that, as motivated in Section 2.2, representation trained with a higher temporal resolution also captures slower processes at the cost of introducing more information into the representation. This can be clearly seen by observing the number of distinct clusters emerging in the visualized representations.

Figure 10 reports the effect of the regularization strength for T-IB representations of the Prinz 2D data. Consistently with the hypothesis, the best-performing model is the one that produces minimal sufficient representations at the training time scale $\tau = 64$. This corresponds to a regularization strength of $\beta = 0.01$.

Figure 11 compares the effects of several regularization strengths, demonstrating the differences between deterministic and stochastic encoders. Deterministic encoders (in yellow) tend to sharply transition from a fully informative representation (on the left) to a constant uninformative representation (on the right). A secondary advantage of using a stochastic encoder is the possibility to compute an upper bound of superfluous information thanks to the expression for the density $p_\theta(\mathbf{z}_t|\mathbf{x}_t)$. This is generally not possible for a deterministic encoder for which $KL(p_\theta(\mathbf{z}_t|\mathbf{x}_t)||q_\phi(\mathbf{z}_t|\mathbf{z}_{t-\tau}))$ can be evaluated only up to a constant. Regularization strength $\beta$ for T-IB is selected based on validation performance: $\beta = 0.01$ for Alanine Dipeptide and Villin; $\beta = 0.001$ for Chignolin.

In our experiments on molecular data, we observed that even small values of $\beta$ can have a substantial impact on reducing the amount of superfluous information contained in the representations, with only a moderate impact on autoinformation. We believe the possible reduction of autoinformation for larger $\beta$ is due to the fact that processes faster than $\tau$ cannot always be fully disentangled. This includes processes that contain lots of information at smaller time scales, but are only marginally informative for events that are $\tau$ time-steps apart. Reducing information regarding faster processes can drop the amount of superfluous information in the representation but still decrease autoinformation whenever the faster process can not be temporally disentangled. Nevertheless, regularization strength in the order of $10^{-3}$ reduces the amount of superfluous information by a substantial factor ($10\times$) with little to no effect on the amount of extracted autoinformation at $\tau$.

Figure 12b shows the effect of the train lag time selection on T-IB models trained with $\beta = 0.01$ and a 2-dimensional representation space for the Villin trajectory. Smaller train lag time corresponds to higher information content and more complex representations, while larger train time scales are associated with simpler representations that are not suitable for unfolding simulation at higher temporal resolution. Note that the larger the training lag-time the longer the training trajectories need to be.

## F.2    AUTOINFORMATION FOR LARGER REPRESENTATIONS

Plots in Figure 4b, Figure 14a, and Figure 13 confirm that with an appropriate regularization strength, T-IB model preserves the maximum amount of autoinformation at the training timescale while decreasing autoinformation for smaller lag times (left of the dashed lines).

Note that the autoinformation plot all the models considered in this analysis matches for large time scales. This suggests that all the corresponding representations preserve autoinformation at large lag times while still differing in the amount of superfluous information at faster scales and the representation structure. The perfect overlap is also justified by Lemma 3 which guarantees that representations that preserve autoinformation at some lag time $\tau$ must also preserve information at larger lag times.

Encoders trained with the VAMPNet objective on complex systems tend to preserve autoinformation only for slower processes. We further observe that VAMPNet models tend to become less numerically stable for increasing representation size, while methods based on non-linear autoinformation maximization are less affected by this hyperparameter choice.

## F.3    EVALUATING STATISTICS FOR MULTIPLE TARGETS AND TIME-STEPS

Figure 15, Figure 14b, and Table 1 report the values of average Jensen-Shannon divergence for transition distribution for different targets $\mathbf{y}_t$. We observe that the T-IB model consistently outperforms the other models for transition matrices computed based on different objectives and several lag times.

One of the main challenges for the evaluation of statistics of slow processes (large transition times in Figure 15) lies in the limited amount of test time frames. We observed that, for large time intervals, the estimation of the ground-truth transition distribution from rare states may be too noisy to produce accurate measures of Jensen-Shannon divergence. As a result, the values reported for large transition times (x-axis) become dependent on the specific test trajectory used for evaluation. Nevertheless, we believe that the relative comparison between the performance of different models may still represent their ability to match the original statistics. More accurate quantitative analysis in this regime would require access to much longer molecular simulations.

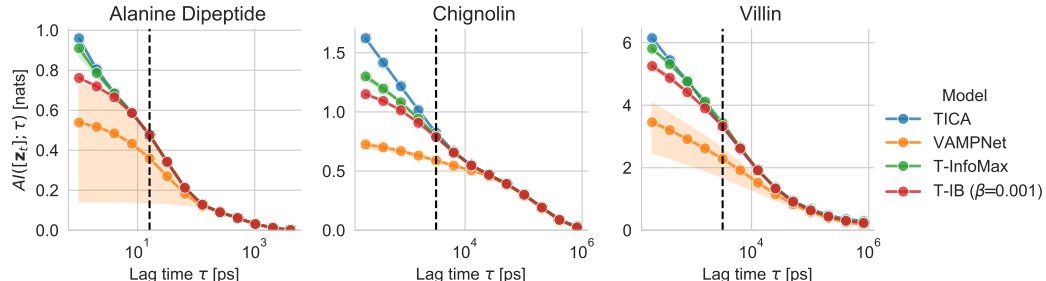

Figure 13: Autoinformation plot for high dimensional representations (16 for Alanine Dipeptide, 32 for Chignolin and Villin). Shaded regions indicate the standard deviation measured across 3 seeds and the dashed vertical line indicates the lag time at which the representations are trained. Representations trained with the VAMPNet objective are generally less consistent (higher variance) across different seeds. T-IB produces sufficient representations (maximal autoinformation at the training time scale) while minimizing the autoinformation for smaller scales.

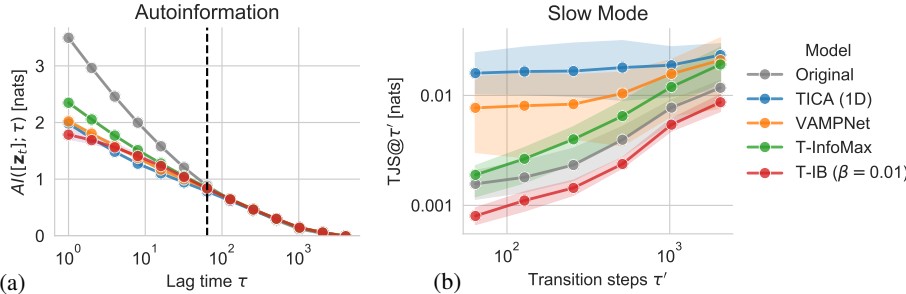

Figure 14: Measurements of autoinformation and transition $JS$ estimated for several time scales. 14a: values of autoinformation estimated at several lag times $\tau$ for representations trained with $\tau = 64$. 14b: values of transition $JS$ estimated at several time scales $\tau'$ from unfolded trajectories. T-IB contains the least autoinformation at small time scales while preserving information at the train lag time or larger. At the same time, T-IB results in the smaller $TJS$ at all the considered time scales. The measure of standard deviation is obtained by considering 3 seeds for each model.

| | Chignolin | | | | Villin | | | |
| | CC Cluster | | RMSD Cluster | | CC Cluster | | RMSD Cluster | |
| | $MJS$ | $TJS$@51.2 ns | $MJS$ | $TJS$@51.2 ns | $MJS$ | $TJS$@51.2 ns | $MJS$ | $TJS$@51.2 ns |
|---|---|---|---|---|---|---|---|---|
| TICA | $7.5 \pm 7.6$ | $5.2 \pm 3.4$ | $7.4 \pm 7.8$ | $6.4 \pm 3.7$ | $1.7 \pm 0.5$ | $6.1 \pm 2.4$ | $7.3 \pm 6.1$ | $5.3 \pm 3.7$ |
| VAMPNet | $30 \pm 20$ | $103 \pm 68$ | $31.9 \pm 21.8$ | $117 \pm 102$ | $63 \pm 88$ | $57 \pm 47$ | $7 \pm 41$ | $40 \pm 7$ |
| T-InfoMax | $5.0 \pm 1.7$ | $3.5 \pm 0.8$ | $4.8 \pm 1.6$ | $3.3 \pm 0.7$ | $2.1 \pm 0.5$ | $5.3 \pm 1.9$ | $5.8 \pm 2.3$ | $8.5 \pm 2.4$ |
| T-IB | $3.3 \pm 2.3$ | $1.1 \pm 0.2$ | $2.9 \pm 2.2$ | $4.1 \pm 1.1$ | $0.8 \pm 0.3$ | $4.4 \pm 1.1$ | $1.7 \pm 1.1$ | $4.1 \pm 1.6$ |

Table 1: Values of marginal ($MJS$) and transition ($TJS$) Jensen-Shannon divergence for trajectories unfolded on latent spaces obtained with different models for the prediction of the CC and RMSD cluster targets described in Appendix E.1.2. The regularized T-IB model consistently outperforms the corresponding unregularized counterpart (T-InfoMax) at the considered time scale.

# G  SIMULATION TIME

## G.1  MOLECULAR DYNAMICS SIMULATION

According to the data reported in Shaw et al. (2021), the 64-node Anton 3 supercomputer can simulate up to 250 microseconds per day for a system consisting of $\sim 10^5$ atoms, which is similar to the total atoms in the Villin and Chignolin simulations. On the other hand, a single A100 GPU can simulate only up to 1.5 microseconds each day. The estimate is based on the data reported in Table III in

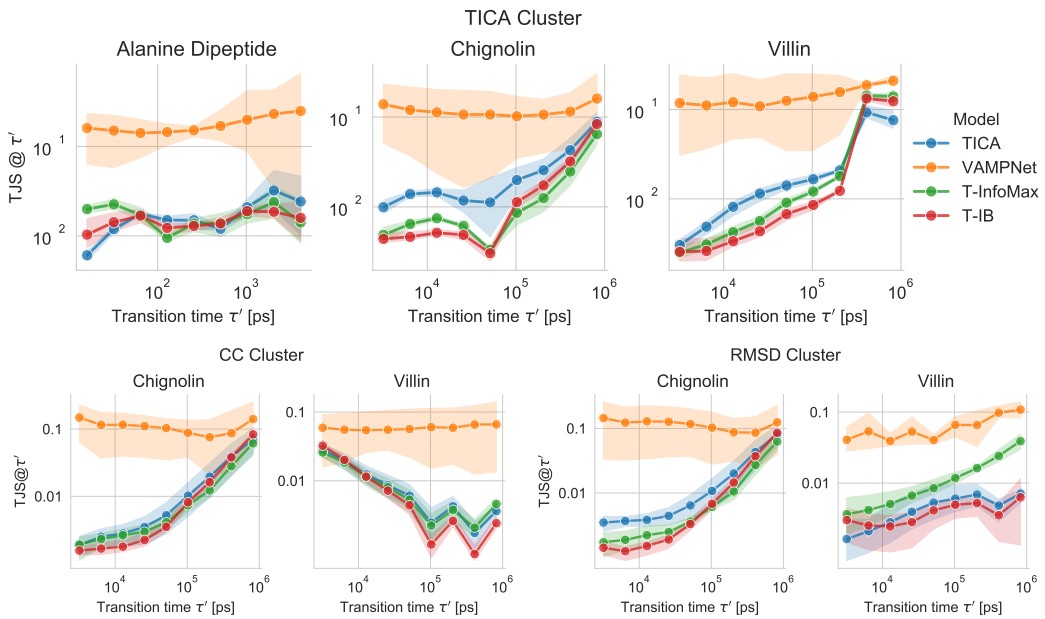

Figure 15: Measure of the average Jensen-Shannon divergence (y-axis) for the unfolded transition matrix for several discrete targets $\mathbf{y}_t$ as a function of the number of unfolding steps (x-axis).

Shaw et al. (2021) and the simulation condition described in the supplementary material provided by Lindorff-Larsen et al. (2011). Therefore, simulating a time jump of $\tau = 3.2$ nanoseconds would require approximately $200s$ on an A100 GPU and about 1 second on Anton 3.

## G.2 LATENT SIMULATION

Unfolding one transition step using the Flow++ transition model used in our experiments requires approximately 100 milliseconds on a single A100 GPU. As a result, for the reported Chignolin and Villin experiments, our estimated acceleration is about a factor $\times 1000$ compared to molecular simulations on the same hardware and $\times 10$ for the highly specialized Anton 3 supercomputer. The total simulation time to produce a new molecular simulation of the same length as the training one ($T \sim 100$ microseconds) is approximately 3 months on a single A100 GPU, 1 hour with Latent Simulation on the same GPU, and 10 hours on 64-nodes Anton 3.

Note that total time require to unfold a latent simulation $T^{LS}$ decreases as we increase the lag time $\tau$:

$$T^{LS} = T\, t^{LS}/\tau,$$

in which the cost $t^{LS}$ is determined by the size of the latent space, the transition model, the prediction model, and the hardware. As shown in Table 2, for our experiments, the prediction time is negligible when compared to the cost of unfolding latent transitions. This is because the prediction model consists of a simple MLP. Whenever the target of interest $\mathbf{y}_t$ is also high-dimensional, the prediction cost may increase significantly. However, it is reasonable to assume both $t_P^{LS}$ and $t_T^{LS}$ to require in the order of 100 milliseconds for most tasks of interest.

The Latent Simulation process is also highly parallelizable. As shown in Table 2, it is possible to simultaneously unfold more than $10^5$ trajectories on a single A100 GPU with little to no overhead.

It is important to note that learning encoder, transition, and prediction models for Latent Simulation still require several ground truth trajectories of length $T >= \tau$, and the unfolded Latent Simulations are approximations of the molecular dynamics. This is because we do not directly represent the water molecules around the proteins nor the single atoms composing the amino acids.

|            | 10 Trajectories | 100 Trajectories | 1000 Trajectories | 10000 Trajectories |
|------------|-----------------|------------------|-------------------|--------------------|
| Transition | $124 \pm 3$     | $128 \pm 1$      | $130 \pm 1$       | $166 \pm 1$        |
| Prediction | $0.690 \pm 0.001$ | $0.700 \pm 0.001$ | $0.732 \pm 0.002$ | $0.739 \pm 0.003$ |

Table 2: Estimations for the time (in milliseconds) required to unfold one step of parallel Latent Simulation of Chignoling and Villin on a single A100 GPU. The estimates refer to the time required to produce samples from the conditional distribution $q_\phi(\mathbf{z}_{t+\tau}|\mathbf{z}_t)$ and $q_\psi(\mathbf{y}_t|\mathbf{z}_t)$ for given $\mathbf{z}_t$. Note that the cost of conditioning and sampling the transition model dominates the one of making predictions. Details on the architectures for the transition prediction models are described in Appendix E.2.

|                    | TICA $[s]$      | VAMPnet $[10^3 s]$ | T-InfoMax $[10^3 s]$ | T-IB $[10^3 s]$ |
|--------------------|-----------------|--------------------|----------------------|-----------------|
| Alanine Dipeptide  | $1.04 \pm 0.01$ | $2.39 \pm 0.02$    | $2.63 \pm 0.08$      | $2.78 \pm 0.02$ |
| Chignolin          | $42.2 \pm 0.2$  | $2.8 \pm 0.3$      | $3.0 \pm 0.3$        | $3.5 \pm 0.3$   |
| Villin             | $60 \pm 1$      | $12.5 \pm 0.3$     | $12.8 \pm 0.2$       | $13.3 \pm 0.4$  |

Table 3: Training time required to train the encoder architectures on the Alanine Dipeptide, Chignolin, and Villin data. The measurements are reported in seconds for the TICA experiments, and $10^3$ seconds for the other models relying on TorchMD encoders.

| Data              | Training Time $[10^3 s]$ |
|-------------------|--------------------------|
| Alanine Dipeptide | $1.7 \pm 0.1$            |
| Chignolin         | $4.5 \pm 0.3$            |
| Villin            | $4.5 \pm 0.5$            |

Table 4: Estimated training time required to fit the transition and predictive model for a fixed representation. The estimates also include the time required to unroll and evaluate latent simulation for validation purposes.

## G.3   TRAINING TIME

We report the training time corresponding to all the models reported in our experimental section by differentiating the time required to train the encoder (step 1) from the training of transition and prediction model (step 2) described in Section 2.1.

Table 3 reports the total time required to train the encoder architectures with the TICA, VAMPnet, T-InfoMax and T-IB objectives. The training time for TICA is substantially shorter since it relies on linear mapping instead of a flexible TorchMD architecture. The variance of the time estimates is computed over three runs per experiment.

The training time for the second step is equivalent for all models since the same transition and prediction architecture are fit to each representation using maximum likelihood. Train time is not influenced by the encoder (linear or Deep NN) since we encode and store the entire dataset to disk at the end of step 1. As a result, the total cost depends solely on the dataset size and size of the latent representation, as reported in Table 4.

The total training time (step 1 + step 2) for the T-IB model on Villin amounts to approximately 5 hours. Unfolding a latent simulation of the same length of the training trajectory requires another hour, bringing the total to 6 hours. Even by including the training time, Latent Simulation is $100\times$ to $1000\times$ faster than running molecular dynamics on comparable hardware.

