# OpenReview forum: "Latent Representation and Simulation of Markov Processes via Time-Lagged Information Bottleneck"
_ICLR.cc/2024/Conference — ICLR 2024 poster_

### Official Review · Reviewer_Yi96 · 2023-10-29

**Soundness:** 3 good
**Presentation:** 3 good
**Contribution:** 3 good
**Rating:** 8
**Confidence:** 2

**Summary:**

The paper aims to solve the problem of accurately and efficiently sampling large-scale systems at long time scales. The solution is based on information theory where the aim is to learn latent representations that capture the dynamical properties of the original process at at a selected time-lag by capturing relevant temporal features and discarding irrelevant high frequency information. The final objective is a trade-off between sufficiency (how informative is the learned representation of the dynamic properties of interest) and minimality of irrelevant information.

**Strengths:**

To the best of my knowledge the paper is original and presents novel ideas. The quality and clarity of the paper overall is also commendable, with well explained notations and good structuring of the paper. The idea is significant since efficiently and accurately simulating large scale systems on long time scales is a critical problem in various fields. The experiments seem to largely support the claims made in the paper.

**Weaknesses:**

This is a thorough work on its own but a few things can be improved, These are detailed in the "Questions" below. Generally this pertains to the way related works section is written and also a few claims that don't seem to be backed up.

**Questions:**

1: The related works section currently reads like a list of related works. It will be far better to re-write this section as a discussion of how current literature relates to this proposed work and how this proposed work builds on top of this existing literature, and how it is better than its closest related works.

2: Can the authors comment on "Since $z_t$ is derived from $x_t$, the autoinformation for $x_t$ sets an upper-bound for the autoinformation for $z_t$. Given that the encoder is time-independent, how can we expect the autoinformation between $x_t$ to be related in any way to the autoinformation in $z_t$?

3: The paper sets up the reader to expect a solution that is faster than current available methods for simulation. I was expecting experiments measuring how long it takes for the simulations to be run compared to existing methods. Can the authors provide that?

If above questions are addressed I'll be happy to re-visit my score.

[UPDATE: authors addressed above points sufficiently. Score has been updated.]

---

> ### Author Response · Authors · 2023-11-20
>
> We express our gratitude to Reviewer Yi96 for their recognition of our work’s value and for the insightful questions and comments, that have prompted updates to both our main text and supplementary material.
>
> 1) **Related work**
>
>    In response to the reviewer’s comment, we have restructured the related work section. This revision aims to more clearly highlight the distinctions between our contributions and existing literature, thereby providing a better context for our work.
>
> 2) **Autoinformation and Data Processing Inequality**
>
>    The assertion in section 2.2 stems from the Data Processing Inequality (Theorem 2.8.1, page 32 in [1]). The underlying intuition is that processing the original data sequence cannot generate new temporal information. A detailed proof for this statement is included in Appendix B.3 and relies on assumption A.1 (appendix B.1), which is based on the graphical model illustrated in Figure 2a. We included a reference to the appendix following the statement in the main text to clarify this.
>
> 3) **Runtime**
>
>    Addressing the reviewers’ comments, we decided to include a detailed analysis and comparison of the simulation runtimes by adding dedicated supplementary material (Appendix G), which is referenced in the experimental section. For further information, we direct the reviewer to our shared answer where these details are elaborated upon.
>
> ### References
> [1] Thomas M. Cover, Joy A. Thomas "Elements of information theory." (1991).

---

> > ### Comment · Reviewer_Yi96 · 2023-11-20
> > **Response**
> >
> > Thank you for making the changes and for the clarification. I believe the paper with these changes, despite the limitations the authors acknowledged, will be a useful addition to published literature on the topic. I'm increasing my score from marginally accept to accept.

---

### Official Review · Reviewer_mH7G · 2023-11-01

**Soundness:** 4 excellent
**Presentation:** 4 excellent
**Contribution:** 4 excellent
**Rating:** 8
**Confidence:** 4

**Summary:**

The paper addresses the challenge of accurately simulating large-scale Markov processes over extended time scales, a computationally expensive task. To overcome this, the authors introduce an inference scheme based on the Time-lagged Information Bottleneck (T-IB) framework. T-IB seeks to simplify the simulation process by mapping complex systems into a reduced representational space that captures relevant temporal features while discarding high-frequency information. Through experiments, the paper demonstrates the efficacy of T-IB in creating information-optimal representations for modeling the statistical properties and dynamics of the original process at a selected time lag.

**Strengths:**

The introduction of the Time-lagged Information Bottleneck (T-IB) framework, rooted in information theory, offers a novel perspective on data-driven model reduction of complex stochastic systems. This framework is developed based on Tiwary et al.'s past-future Information Bottleneck (IB) theory and leverages recent advancements in machine learning to achieve improved estimation of time-lagged mutual information. The experiments showcases the robustness and effectiveness of T-IB in learning information-optimal representations for Markov process simulation.

**Weaknesses:**

1. Comparison with Past-Future IB Method: The paper builds upon the past-future IB method proposed by Tiwary et al., but it introduces an optimization objective for I(z_t, z_t+tau) instead of I(z_t, x_t+tau). While theoretically, optimizing I(z_t, z_t+tau) is expected to yield a lower bound on I(z_t, x_t+tau), the paper lacks a comprehensive discussion of the differences between the two approaches. A deeper exploration of the distinctions and the implications of this choice is needed to provide a clearer understanding.

2. Choice of Estimation Method: In the machine learning domain, there are various methods for estimating lower bounds on mutual information. The paper selects a specific method, but it would benefit from a more thorough justification for this choice. Providing a comparative analysis of the selected method against other available options would enhance the paper's robustness.

3. Complexity of Additional Terms in Formula (8): The introduction of two additional terms in Formula (8) may appear somewhat perplexing. From an information bottleneck theory perspective, the goal is to discard as much irrelevant information as possible. However, dimensionality reduction inherently leads to information loss. The rationale and impact of introducing these terms need further clarification. Moreover, the experimental results suggest that the inclusion of these terms does not significantly affect the dimensionality reduction outcome, requiring a more detailed explanation.

4. Quantitative Evaluation of Dimensionality Reduction: The paper lacks a robust quantitative evaluation of the dimensionality reduction results. While the field may lack a general method for quantitatively assessing dimensionality reduction outcomes, the authors could consider exploring quantitative evaluation approaches, such as measuring mutual information under different tau conditions, to provide a more concrete assessment of the method's performance.

5. Comparison with VAMPnet: VAMPnet, as mentioned in the weaknesses, primarily serves the purpose of extracting dominant or slow components and estimating kinematic properties. It may not be a direct dimensionality reduction method. Comparing VAMPnet with the proposed approach for dimensionality reduction may not be entirely fair or informative.

**Questions:**

See Weaknesses

---

> ### Author Response · Authors · 2023-11-20
>
> We thank reviewer mH7G for their acknowledgment of the novelty of our work and for providing insightful comments, which we address as follows:
>
> 1) **Lack of comprehensive discussion regarding the mutual information lower-bound and estimation method**
>
>    The decision to employ a contrastive bound on $I(z_t;z_{t+\tau})$ instead of $I(z_t;x_{t+\tau})$ was driven by several factors:
>
>       * *Computational efficiency*: $z_{t+\tau}$ is usually much lower-dimensional than $x_{t+\tau}$. Using contrastive learning to maximize information between representations becomes computationally more efficient than employing a decoder to predict $x_{t+\tau}$.
>
>
>       * *Additional Independence Condition*: the condition (i) $I(z_t;z_{t+\tau})=I(x_t;x_{t+\tau})$ is stronger than (ii) $I(z_t;x_{t+\tau})=I(x_t;x_{t+\tau})$ since it requires $I(x_t;x_{t+\tau}|z_{t+\tau})=0$, which is not directly implied by (ii). This additional independence is crucial in our Lemma 3 (refer to Appendix B.7), providing a stronger foundation for our theoretical framework. Although the results from Lemma 1 and 2 can be extended under condition (ii), the specific independence in (i) plays an important role in our methodology.
>
>
>       * *Successful examples*: Numerous contemporary studies have successfully employed contrastive methods to maximize information among representations, especially in handling structured, high-dimensional data sequences [1,2]. This contrasts with the limitations observed in reconstruction-based mutual information maximization methods, particularly in the context of molecular dynamics, as highlighted by previous research [3].
>
>
> 2) **Explicit superfluous information minimization vs Dimensionality Reduction**
>
>    We appreciate the reviewer's insights regarding dimensionality reduction and its relationship with information loss. In response, we'd like to elaborate on several pertinent aspects:
>
>
>       * *Non-Guaranteed Compression*: Reducing the dimensionality of the data does not necessarily result in compression. In cases where high-dimensional data resides on a lower-dimensional manifold, non-linear encoding methods have the potential to map data into a reduced dimensionality without incurring information loss.
>
>       * *Generalization*: Contemporary research on generalization bounds in representation-based prediction underscores the significance of superfluous information measures [4]. These bounds are more directly related to the minimization of irrelevant information rather than solely focusing on the dimensionality of the data. This perspective aligns with our approach, as we emphasize the explicit reduction of superfluous information.
>
>
>       * *Applicability*: When dealing with low-dimensional data, such as the Prinz-2D dataset depicted in Figure 3, dimensionality reduction may not be a feasible or a necessary strategy. However, our approach to minimizing superfluous information retains its applicability and effectiveness even in these scenarios. By focusing on the reduction of non-essential information, our methods can still contribute to the compression and enhanced understanding of the data without necessarily reducing its dimensionality.
>
> 3) **Quantitative Evaluation of Dimensionality Reduction**
>
>    We quantitatively assess representations by examining the autoinformation at various timescales, as depicted in Figures 4b, 12, and 13.  Additionally, we measure the amount of information carried about relevant $y^{slow}$ and irrelevant $y^{fast}$ features (Figure 3b).
>    Despite the subtlety of qualitative differences in Figure 4a, a consistent decrease in information at faster timescales is observed across Figures 4b and 12. Figure 11 offers a visual representation of how different $\tau$ settings impact the representations of the Villin dataset, further clarifying the effects of our information reduction approach.
>
>
>    While our method indeed facilitates dimensionality reduction, it's important to emphasize that our primary objective is to leverage the representation for latent simulations. Consequently, we assert that evaluating the statistical fidelity of simulations unrolled from each representation provides a meaningful metric for assessing representation quality. This evaluation becomes particularly significant considering that the transition and prediction models, along with the training procedures, remain constant across all fixed representations.

---

> ### Author Response · Authors · 2023-11-20
>
> 4) **Comparison with VAMPnet**
>
>
>    The main purpose of including the VAMPnet visualization in Figures 3c and 4a is to provide intuitive insights regarding the effect of the training objective and encoder choice on the resulting representation. More specifically, comparing the TICA and VAMPnet representations showcases the effect of choosing a linear vs non-linear encoder. Furthermore, Since VAMPnet employs the same encoder architecture as the T-InfoMax model, this comparison also visually emphasizes the differences between maximizing linear and non-linear autoinformation. This aspect is particularly significant, as it helps in comprehending the subtle yet impactful distinctions brought about by different maximization strategies.
>
>    It is important to note that, following previous research [5], the quantitative evaluations of the fidelity of the unfolded latent simulations for VAMPnet refer to higher-dimensional representations (32 dimensions).
>
>
> ### References
> [1] Oord, Aaron van den, Yazhe Li, and Oriol Vinyals. "Representation learning with contrastive predictive coding." arXiv preprint arXiv:1807.03748 (2018).
>
> [2] Radford, Alec, et al. "Learning transferable visual models from natural language supervision." International conference on machine learning. PMLR, 2021.
>
> [3] Chen, Wei, Hythem Sidky, and Andrew L. Ferguson. "Capabilities and limitations of time-lagged autoencoders for slow mode discovery in dynamical systems." The Journal of Chemical Physics 151.6 (2019).
>
> [4] Kawaguchi, Kenji, et al. "How Does Information Bottleneck Help Deep Learning?." arXiv preprint arXiv:2305.18887 (2023).
>
> [5] Sidky, Hythem, Wei Chen, and Andrew L. Ferguson. "Molecular latent space simulators." Chemical Science 11.35 (2020): 9459-9467.

---

> ### Comment · Reviewer_mH7G · 2023-11-21
>
> Thank authors for the reply. Some more questions and comments:
>
> 1) For the difference between $I(z_t, z_{t+\tau})$ and $I(z_t,x_{t+\tau})$, I agree that the former one can be estimated more efficiently. But under the assumption $I(z_t,x_{t+\tau})=I(x_t,x_{t+\tau})$, we can also obtain the conclusion $I(z_t,x_{t+\tau'})=I(x_t,x_{t+\tau'})$ for $\tau'>\tau$, which is similar to Lemma 3.
>
> 2) In Figure 4b, it seems that there is a significant reduction in autoinformation when $\beta=0.1$ compared to $\beta=0$. However, the authors show the results for $\beta=0.1$ in Figure 4a. I am curious to understand the rationale behind the authors' choice of $\beta$. What criteria or methodology guided the selection of $\beta$?
>
> 3) I would like to reemphasize that VAMPnet is a method for estimating dominant spectral components. In other words, the more accurately the model estimates, the more precisely it forgets those fast components. Therefore, I do not consider that it is reasonable to compare the AIG of VAMPnet with other methods. Authors should highlight this point in the analysis of experiments.

---

> > ### Author Response · Authors · 2023-11-21
> >
> > We thank the reviewer for their engagement and insightful comments.
> >
> > 1) Regarding the extension of Lemma 3, we concur with the reviewer's observation. We realized that it is indeed feasible to extend the result of Lemma 3 to the condition $I(z_t;x_{t+\tau})=I(x_t;x_{t+\tau})$. This oversight was due to a difference in our proof approach, but we have now recognized that Lemma 3 can be straightforwardly extended by leveraging the Markov property $x_t\to x_{t+\tau}\to x_{t+\tau'}$. We are adding this discussion to Appendix D to provide a clearer comparison and differentiation between the two approaches.
> > 2) The selection of $\beta$ values in our experiments was based on the validation performance from the set $\\{10^{-5}, 10^{-4}, \dots, 10^{0}\\}$. We specifically showcased the representations obtained with $\beta=0$ (T-InfoMax) and a stronger regularization ($\beta=0.1$) to illustrate the impact of the regularization term more distinctly. A detailed evaluation exploring the influence of different regularization strengths can be found in Appendix F.1, which provides a deeper understanding of how $\beta$ affects our model's performance.
> > 3) Regarding the measurement of AIG for VAMPnet representations, our primary goal was to gain insights into the time scale of the information captured by the representations. Given that VAMPnet representations have been utilized for latent simulation [5], and considering that the latent simulation error is upper-bounded by the AIG at the simulation scale $\tau$ (Equation 5), we believed that including this measurement would provide meaningful insights. However, we recognize and agree with the reviewer's point about the primary purpose of the VAMPnet model. Consequently, we are updating the main text to emphasize the fundamental goal of the VAMPnet model and its implications for our analysis.

---

> > > ### Comment · Reviewer_mH7G · 2023-11-23
> > >
> > > Thank authors for the detailed reply.

---

### Official Review · Reviewer_H2G8 · 2023-11-01

**Soundness:** 3 good
**Presentation:** 3 good
**Contribution:** 3 good
**Rating:** 8
**Confidence:** 2

**Summary:**

Markov processes can require computing many intermediate steps before reaching the final state which may be expensive. This paper proposes modeling state in latent space and computing large steps in time directly. The transition and emission distributions are approximated using variational inference. The optimization scheme is two step: first the encoding distribution $q(z | x)$ is learned and then the rest is learned. The approach ensures that the latent space contains information about the original state through the mutual information and "autoinformation". They introduce a loss T-InfoMax that maximizes mutual information on latent state across some resolution gap. Finally, they introduce an information bottleneck to discard unnecessary information such as time dependent information, resulting in a loss T-IB. All the mutual information terms are approximated, some are learned with neural contrastive estimation.

**Strengths:**

The paper is nicely written and the method is explained well.

The information theoretic approach is a good motivator for what follows in the paper. The method itself is going through a couple of departures from the pure mutual information and get increasingly complex and further approximated.

The theoretical results in 2.2 are nice and follow the story, although I did not check the proofs in detail.

The results look good although some additional discussion is missing.

**Weaknesses:**

The method has many moving parts and some choices are not well motivated. For example, different choices of architectures for different distributions, training with contrastive estimation. A study that sees which part is contributing to the performance would be nice.

The training requires two step procedure.

As far I can see the authors do not provide actual run times. What should be compared is the total time to obtain the training data, train the model and perform the simulation.

There are two experiments, one with synthetic data which is a toy example and one with molecular simulation which again seems small in scale.

**Questions:**

- How to choose $\tau$ and once chosen, does this parameter remain fixed and the model has to be retrained for different choice?

- Can you provide run times?

- How does the model scale with the amount of data available for training?

---

> ### Author Response · Authors · 2023-11-20
>
> We are grateful to the reviewer for their positive remarks about our presentation and for the insightful comments and questions raised.
>
> To address the main points:
>
> 1) **Architectural choices**
>
>    We acknowledge that the main text could have provided more comprehensive details regarding the selection of encoder, transition, prediction, and mutual information maximization models. Due to the constraints of space, our focus was primarily on delineating a general family of models, directing readers to relevant literature for specifics on architectural choices. Our architecture selection was primarily driven by two key principles:
>
>       * *Flexibility*: Our primary criterion was to choose architectures known for their effectiveness in modeling similar problems. This led us to opt for contrastive learning methods over reconstruction-based methods for mutual information maximization [1], use TorchMD encoders for invariant molecular representations, and select advanced normalizing flows like Flow++ for transitions, instead of simpler mixture models.
>
>
>       * *Computational cost*: Another important consideration was computational efficiency. While we acknowledge that more flexible transition models like diffusion models might yield more precise transitions, they could also slow down latent simulations. Exploring this trade-off is an exciting avenue for future research.
>
>    We recognize that the performance of our models could potentially be enhanced with more nuanced choices for the various architectures. However, it is crucial to emphasize that the primary objectives of this submission are to establish a formal foundation for the T-InfoMax and T-IB losses and to demonstrate the efficacy of our proposed inference and representation learning method. We aim to show that, with reasonable architectural choices, our approach compares favorably against different training objectives using the same architectures.
>
>
> 2) **How to choose $\tau$**
>
>
>    The selection of $\tau$ essentially serves as a task-specific hyper-parameter. For instance, in weather prediction, $\tau$ might be set to one day for daily forecasts, but could be reduced for predictions requiring higher temporal resolution, such as hourly rain forecasts. Similarly, in molecular dynamics, the time scale for $\tau$ can be aligned with the expected duration of folding/unfolding events or other significant state transitions.
>
>    Figure 11 in Appendix F illustrates the impact of varying $\tau$ on the representation and the corresponding autoinformation values. A smaller $\tau$ tends to retain more informative details, capturing faster processes and a greater number of clusters.  Conversely, a larger $\tau$ emphasizes slower processes, discarding fast dynamics and speeding up the latent simulation process due to the larger temporal jumps.
>
>    An important consideration in choosing $\tau$ is the necessity of having multiple training examples with pairs spaced $\tau$ apart. Consequently, the selection of $\tau$ is often bounded by the feasibility of simulating trajectory lengths greater than $\tau$. In our experiments, we opted for a $\tau$ that is several orders of magnitude shorter than the total trajectory length. However, we propose that the total simulation time $T$ of available trajectories need not be significantly larger than $\tau$, especially if multiple parallel simulations are utilized instead of a single long trajectory. This approach is particularly relevant in the context of molecular dynamics since producing long ground truth trajectories is computationally intensive. Exploring this efficiency in leveraging parallel trajectories represents a promising direction for future research.
>
>
> 3) **Does this parameter remain fixed?**
>
>    Once an encoder is trained with T-IB or T-InfoMax for a given $\tau$, it is possible to learn transition models $ q_\phi(z_t|z_{t-\tau’}) $ for any larger $\tau’\ge\tau$. This implies that we can retrain the transition model for a larger temporal scale $\tau’>\tau$ without needing to modify the encoder or the predictor $q_\psi(y_t|z_t)$.
>
>    However, with T-IB, it's often advantageous to retrain or fine-tune the encoder for a new lag time $\tau’$. This adjustment allows the encoder to disregard processes slower than $\tau’$, thereby simplifying the representation. When updating the encoder, it's necessary to retrain both the transition and predictive models as well.
>
>    Conversely, reducing the time-scale $\tau’<\tau$ necessarily requires re-training the encoder to incorporate faster processes. Utilizing previously trained encoder, transition, and prediction models as starting points should speed up the training.
>
>
> 4) **Can you provide run times?**
>
>    We thank the reviewer for highlighting the need for runtime measurements. We have now included estimates for Molecular Dynamics runtime in Appendix G.1, Latent Simulation runtime in G.2, and training time in G.3. These estimations have been referenced in the main text.

---

> ### Author Response · Authors · 2023-11-20
>
> 5) **How does the model scale with the amount of data available for training?**
>
>
>    In our experiments, due to data availability constraints, we focused on a single long simulation for each molecule, exploring all relevant states and approaching equilibrium distribution.  We experimented with lag times up to 204.8 ns, yielding meaningful molecular representations. We believe that longer training trajectories could enable us to extend the time scale further. Additionally, using more training trajectories could improve the accuracy of our autoinformation estimates, as discussed in Appendix C.2. Enhanced autoinformation estimates would not only allow the model to more effectively capture features at the chosen time scale but also provide more precise quantitative estimations, such as those shown in Figure 4, aiding in diagnosis and evaluation.
>
>
>
>
> ### References
> [1] Chen, Wei, Hythem Sidky, and Andrew L. Ferguson. "Capabilities and limitations of time-lagged autoencoders for slow mode discovery in dynamical systems." The Journal of Chemical Physics 151.6 (2019).

---

> > ### Comment · Reviewer_H2G8 · 2023-11-22
> >
> > Thank you for your clarifications. I'm updating my score accordingly.

---

### Author Response · Authors · 2023-11-20
**Shared answer**

We thank the reviewers for their insightful comments, questions, and suggestions. We will start by addressing the question about the Latent Simulation runtime that was asked by multiple reviewers.



## Runtime and Applicability
In response to the query regarding the effective runtime of our proposed methods, we have added estimates and measurements of both simulation and training times in Appendix G, which is now referenced in the experimental section.

We first acknowledge a crucial limitation of our current approach: the generation of initial training data through computationally expensive molecular dynamics (MD) simulations. This initial data-generation process is not only time-consuming but also constrains the general applicability of our model when trained on a single protein system. While our method is capable of generating novel simulations that exceed the length of the training data, it's important to recognize that without the ability to generalize to unseen systems or relying solely on short trajectories, our model is not yet a practical replacement for traditional MD simulations.

Despite this limitation, we measured that training the T-IB model and unfolding a Latent Simulation for a duration equivalent to the training data approximately takes 6 hours—5 for training and 1 for unfolding. This contrasts sharply with the up to 3 months required for performing molecular dynamics (MD) simulations on the same hardware (one A100 GPU), highlighting a substantial speed-up of about 1000x. Additionally, Latent Simulations are highly parallelizable. We observed that a single A100 GPU can efficiently unfold over 10,000 latent simulations simultaneously with minimal overhead.
Furthermore, Latent Simulations can be further accelerated by considering larger values of $\tau$, although this would require longer trajectories during training.

Looking ahead, future research that assesses the generalization capabilities of T-InfoMax and T-IB Latent Simulations could greatly enhance the practicality of latent simulation in molecular dynamics (MD). As our runtime assessments have demonstrated, the latent simulation procedure has the potential to significantly accelerate the simulation process. Additionally, we recognize that our method could be directly applicable to systems with extensive historical data, such as weather forecasting or ocean temperature prediction. However, we decided to showcase the foundational efficacy of our approach within the realm of MD, given the extensive literature, its specific challenges, and its complexity. The exploration of these additional applications, while promising, remains an area for future investigation beyond the scope of this work.

---

### Meta-Review · Area_Chair_WjQ4 · 2023-12-04

**Metareview:**

This paper proposes a new technique to simplify the simulation of Markov processes using an Information Bottleneck principle. The method is based on mapping into a simpler space, where they use information theoretic techniques to capture the relevant "frequencies". This is more general than some existing methods that work based on assumptions of limited events in short time steps. The reviewers are in agreement on the merits of the paper, though some found it confusing with many moving parts. The reviewers acknowledge that many of their concerns have been clarified since the rebuttal

**Justification For Why Not Higher Score:**

Thorough and solid paper on a specific problem

**Justification For Why Not Lower Score:**

Clearly should be a accepted

---

### Decision · Program_Chairs · 2024-01-16

Accept (poster)